# Evaluating and Learning Optimal Dynamic Treatment Regimes under Truncation by Death

**Sihyung Park**
Department of Statistics
North Carolina State University
Raleigh, NC 27695
spark52@ncsu.edu

**Wenbin Lu**
Department of Statistics
North Carolina State University
Raleigh, NC 27695
wlu4@ncsu.edu

**Shu Yang**
Department of Statistics
North Carolina State University
Raleigh, NC 27695
syang24@ncsu.edu

## Abstract

Truncation by death, a prevalent challenge in critical care, renders traditional dynamic treatment regime (DTR) evaluation inapplicable due to ill-defined potential outcomes. We introduce a principal stratification-based method, focusing on the always-survivor value function. We derive a semiparametrically efficient, multiply robust estimator for multi-stage DTRs, demonstrating its robustness and efficiency. Empirical validation and an application to electronic health records showcase its utility for personalized treatment optimization.

## 1 Introduction

The implementation of evidence-based treatment strategies has grown in importance in healthcare, with dynamic treatment regimes (DTRs; Robins, 1997) being a key component. DTRs can identify optimal treatment strategies even when data is collected from suboptimal policies. Extending beyond traditional static decision rules, DTRs offer personalized, multi-stage treatment sequences that adapt to evolving patient characteristics and treatment histories. Crucially, they facilitate individualized care by determining the right interventions at the right time, tailored to each patient's needs. This personalized approach is essential, particularly in chronic conditions, e.g., alcohol and drug abuse (Murphy et al., 2007), AIDS (Robins et al., 1989), cancer (Zhao et al., 2009), diabetes (Chakraborty and Murphy, 2014), that are characterized by their prolonged duration, often involving repeated cycles of remission and exacerbation and necessitating ongoing medical intervention. Early works proposed g-computation (Robins, 1997), inverse probability weighting (IPW; Robins et al., 2000), Q-learning (Watkins and Dayan, 1992) and A-learning (Murphy, 2003) to evaluate average clinical benefits – value functions – of treatment policies, while more recent literature employs doubly robust forms of value search methods (Zhang et al., 2013) and outcome weighted learning (Liu et al., 2018) to mitigate error due to potential model misspecification.

While statistical methods for estimating optimal DTRs have proliferated, their applicability is often limited by the requirement of well-defined potential outcomes. In studies, truncation by death is a common challenge, which arises when death precludes subsequent data collection thereby leaving outcomes of interest undefined. For example, 15% of sepsis patients in the Medical Information Mart for Intensive Care III (MIMIC-III) database experience death, causing the outcomes at the final 48-hour time point undefined (Rhodes et al., 2024). For individuals underwent truncation by death,

the counterfactual outcomes are not simply missing but ill-defined. This distinguishes such cases from conventional missingness, as it is generally not appropriate to impute an outcome value when it is inherently tethered to a mortality event. Moreover, truncation by death creates non-comparable treatment groups because individuals who survived with treatment may have died without it, thus rendering the population-level value function ill-defined (Rubin, 2006).

Despite its fundamental difference, estimators designed for censoring, such as inverse probability of censoring weighting (Robins and Finkelstein, 2000), are misapplied in practice to address truncation by death. While utilizing time-to-death as the primary endpoint and constructing optimal DTRs to maximize survival-related value functions, e.g., $t$-year survival probability (Jiang et al., 2017) or restricted mean survival time (Rhodes et al., 2024), is another prevalent strategy, there are situations where improving quality of life, functional status, or disease-specific symptoms takes precedence. In these cases, existing approach may potentially neglect patient well-being. Therefore, specialized methods designed to capture treatment effects beyond survival that explicitly account for truncation by death are essential for obtaining desired treatment strategies.

To overcome the difficulties, principal stratification (Imbens and Angrist, 1994, Baker and Lindeman, 1994) provides a valuable approach. Instead of estimating treatment effects across the entire population, this method concentrates on the always-survivor stratum, a latent subgroup characterized by survival under all treatment assignments. By focusing on the always-survivor stratum, which is not affected by truncation, we ensure that the potential outcomes and consequently the associated value function are well-defined. Techniques for identifying the always-survivor value function are presented in single-decision contexts (Chu et al., 2023, Grossi et al., 2025); however, extending these methods to multi-stage DTRs presents significant challenges, as they require navigating complex decision sequences, managing time-varying confounding, and modeling delayed or cumulative treatment effects under possibly correlated variables across time points.

**Contributions** This research introduces a methodology for estimating the always-survivor value function for optimal multiple-decision DTRs. As a foundational step, we introduce a theoretical framework for applying principal stratification to multi-stage decision problems. Based on this framework, our methodology defines a well-defined estimand under truncation by death and proposes an efficient and robust method to estimate it. Our contributions are as follows.

1. We define the always-survivor value function, which is well-defined and identifiable from the observed data (Section 3).

2. We derive the efficient influence function and semiparametric efficiency bound of the estimand. Based on these results, we propose a multiply robust (MR), locally efficient estimator for always-survivor value function (Section 4). Section 5 demonstrates multiply robust off-policy learning using the proposed estimator.

3. We empirically validate the theoretical properties of MR estimator in various nuisance model specification scenarios (Section 6). Across settings, the MR estimator consistently demonstrates robustness to nuisance model misspecification. We show MR estimator can facilitate decision-making for high-risk patients group by applying it to MIMIC-III database (Section 7).

Finally, we refer the reader to the Appendix for more details on proof of theorecial results (Section A.1) and technical description (Section A.2).

## 2 Background and related work

**Notation** Let $k = 1, \ldots, K$ denote the sequence of decision points, and let $Z_k$ represent the generic observed variable of interest in our setting at time point $k$. Following standard notation in dynamic treatment regimes, let $X_k$ be the pre-treatment covariates observed at time $k$, and $A_k$ be the treatment assignment at time $k$, made after observing $X_k$. Let $C_k$ and $S_k$ denote the censoring and survival indicators at time $k$, respectively. If a trajectory experiences dropout after observing $X_k$, we set $C_k = 1$ and consider $S_k$ and all variables at time points $(k + 1)$ and beyond as unobserved. Otherwise, we set $C_k = 0$, and $S_k$ is observable. If a trajectory experiences death at time $k$, we set $S_k = 0$, and all variables at time points greater than $k$ are unobserved. Otherwise, we set $S_k = 1$, and the variables at the next stage are observed. These definitions ensure that the data are subject

to monotone censoring and truncation by death. After $K$ treatments, we observe the outcome $Y$ if no censoring or death event occurred. Without loss of generality, we assume that larger values of $Y$ indicate better outcomes.

We utilize the potential outcome framework to define counterfactuals. Bar notation is used to denote a vector of variables up to a certain time. For instance, the sequence of treatment assignments up to time $k$ is represented as $\bar{a}_k = (a_1, \ldots, a_k)$. We omit the subscript when $k = K$. We denote $Z_k^{\bar{a}_k} = Z_k^{a_1 \cdots a_k}$ as the potential outcome of $Z_k$ had it received treatments $a_1$ through $a_k$. Define $H_k = \{\bar{X}_k, \bar{A}_{k-1}\}$, and let $\mathbf{0}_k$ and $\mathbf{1}_k$ be zero and one-vector of dimension $k$, respectively.

To simplify the methodological development, we focus on the case with $K = 2$ decision points, noting that the results generalize readily to multiple decision points. The observed data are then represented as: $O = \{X_1, A_1, C_1, (1 - C_1)S_1, (1 - C_1)S_1 X_2, (1 - C_1)S_1 A_2, (1 - C_1)S_1 C_2, (1 - C_1)(1 - C_2)S_1 S_2, (1 - C_1)(1 - C_2)S_1 S_2 Y\}$. We assume causal consistency, a standard assumption in causal inference, which implies consistent treatment effects on each unit and no interference between units. For example, since the severity of Sepsis is unlikely to be transmitted or influenced between patients, causal consistency is expected to hold in this case.

**Assumption 1** (Causal consistency).  $Z_1 = Z_1^{A_1}$ and $Z_2 = Z_2^{A_1 A_2}$.

For the value function to be identifiable, sequential randomization is typically assumed. Assumption 2 states this by ensuring the absence of unobserved confounding at each time point in treatment assignments.

**Assumption 2** (Sequential randomization).  $\{Y^{a_1 a_2}, X_2^{a_1}\} \perp\!\!\!\perp A_k \mid H_k$ for all $a_k$, $k = 1, 2$.

Define the deterministic two-stage treatment policy $\pi = (\pi_1, \pi_2)$, where $\pi_1 : \mathcal{X}_1 \to \mathcal{A} = \{0, 1\}$ maps the space of baseline covariates to the treatment space, and $\pi_2 : \mathcal{H} \to \mathcal{A}$ maps the space of variables $\{X_1, A_1, X_2\}$ available at the second stage to the treatment space. For simplicity, let $\pi(\bar{x}) = (\pi_1(X_1), \pi_2(H_2^\pi))$ denote the treatment assignments consistent with the policy, where $Z_1^\pi = Z_1^{\pi_1(X_1)}$ and $Z_2^\pi = Z_2^{\pi(\bar{X})}$ are the potential outcomes of $Z$ consistent with $\pi$.

Dynamic treatment regimes traditionally aim to find the optimal policy $\pi^*$ that maximizes $V(\pi) = \mathbb{E}[Y^\pi]$, the expected outcome under $\pi$, subject to Assumptions 1, 2, and treatment assignment positivity.

**Principal stratification**  Principal stratification classifies data into distinct latent strata defined by a principal stratification variable $U$, which is a combination of post-treatment counterfactuals. Among these strata, the always-survivor stratum is characterized by survival under all treatment assignments. Since this stratum is not affected by death, potential outcomes and, consequently, the value function are well-defined within it.

While principal stratification is a popular method for addressing treatment compliance and truncation by death (Jiang et al., 2022), its traditional framework is limited to single-decision point problems. Chu et al. (2023) proposed a multiply robust estimator for the always-survivor value under censoring and truncation by death; yet, their approach is inherently designed for single-decision scenarios and does not account for the complex time-varying correlation structure inherent in DTRs. Grossi et al. (2025) introduced principal stratification approaches that incorporate longitudinal survival indicators. However, their frameworks remain restricted to a single decision point immediately following the baseline.

We extend principal stratification to multiple decision points, classifying data into latent groups based on the final-stage survival indicator $S_2$. Let $U = (S_2^{00}, S_2^{01}, S_2^{10}, S_2^{11})$ that results in sixteen latent strata, with notable groups described in Table 1. Our focus is on evaluating and optimizing policies using the always-survivor value function $V_{\mathrm{AS}}(\pi) = \mathbb{E}[Y^\pi | U = 1111]$, rather than the traditional value function $\mathbb{E}[Y^\pi]$ that is ill-defined in the presence of death events.

Since principal strata are defined using counterfactuals, they are inherently latent. Therefore, it is essential to identify the always-survivor value function from observed data. We adopt monotonicity and principal ignorability assumptions, following Jiang et al. (2022).

**Assumption 3** (Monotonicity).  $S_2^{11} \geq S_2^{10}, S_2^{11} \geq S_2^{01}, S_2^{10} \geq S_2^{00}, S_2^{01} \geq S_2^{00}, S_1^1 \geq S_1^0, S_1^{a_1} \geq S_2^{a_1 a_2}$, and $C_2^{a_1 a_2} \geq C_1^{a_1}$ almost surely.

**Assumption 4.**  $\mathbb{E}[g(X_2^{a_1}) | X_1, \bar{S}^{a_1 a_2} = \mathbf{1}_2] = \mathbb{E}[g(X_2^{a_1}) | X_1, S_1^{a_1} = 1]$ for an integrable g.

Table 1: Description of Selected Principal Strata.

| $U$ | Stratum type | Description |
|---|---|---|
| 1111 | Always-survivor | Patients who would survive regardless of the treatment assignments |
| 0111 | Protectable | Patients who would survive if treated in either decision points, die otherwise. |
| $\vdots$ | $\vdots$ | $\vdots$ |
| 1000 | Defier | Patients who would die if treated in either decision points, survive otherwise. |
| 0000 | Never-survivor | Patients who would die regardless of the treatment assignments. |

Table 2: List of notation.

| Symbol | Description |
|---|---|
| $\pi$ | A deterministic treatment policy. |
| $X_k$ | Baseline (if $k=1$) or intermediate covariates (if $k=2,\ldots,K$). |
| $A_k$ | Treatment indicator at $k$. 1 for treatment, 0 for control. |
| $C_k$ | Censorship indicator at $k$. 1 for missing, 0 for observed. |
| $S_k$ | Survival indicator at $k$. 1 for survival, 0 for death. |
| $Y$ | Outcome of interest. |
| $H_k$ | History up to $k$. i.e., $\{\bar{X}_k, \bar{A}_{k-1}\}$. |
| $U$ | Principal stratification variable. i.e., $S_2^{00} S_2^{01} S_2^{10} S_2^{11}$. |
| $e_k^{\bar{a}_k}(\bar{x}_k)$ | Propensity score at $k$ defined as $\mathbb{P}(A_k = a_k \mid \bar{x}_k, \bar{a}_{k-1}\bar{C}_{k-1} = \mathbf{0}_{k-1}, \bar{S}_{k-1} = \mathbf{1}_{k-1})$. |
| $c_k^{\bar{a}_k}(\bar{x}_k)$ | Censoring probability at $k$ defined as $\mathbb{P}(C_k = 0 \mid \bar{x}_k, \bar{a}_k, \bar{C}_{k-1} = \mathbf{0}_{k-1}, \bar{S}_{k-1} = \mathbf{1}_{k-1})$. |
| $p_k^{\bar{a}_k}(\bar{x}_k)$ | Survival probability at $k$ defined as $\mathbb{P}(S_k = 1 \mid \bar{x}_k, \bar{a}_k, \bar{C}_k = \mathbf{0}_k, \bar{S}_{k-1} = \mathbf{1}_{k-1})$. |
| $\varphi_k^{\bar{a}_k}(\bar{x}_k)$ | Joint propensity-censoring probability at $k$ defined as $e_k^{\bar{a}_k}(\bar{x}_k) c_k^{\bar{a}_k}(\bar{x}_k)$. |
| $\mu_2^{a_1 a_2}(\bar{X})$ | Outcome regression model defined as $\mathbb{E}[Y|\bar{X}, (a_1, a_2), \bar{C} = (0,0), \bar{S} = (1,1)]$. |
| $m_{\mu_2}^{a_1 a_2}(X_1)$ | Outcome regression model at $k=1$ defined as $\mathbb{E}[\mu_2^{a_1 a_2}(\bar{X})\mathbf{1}\{A_2 = a_2\}|X_1, a_1, C_1 = 0, S_1 = 1]$. |
| $m_{p_2}^{a_1 a_2}(X_1)$ | Eventual survival probability at $k=1$ defined as $\mathbb{E}[p_2^{a_1 a_2}(\bar{X})|X_1, a_1, C_1 = 0, S_1 = 1]$. |

**Assumption 5** (Principal ignorability). *For $u_1, u_2, u_3 \in \{0,1\}$,*

*(i)* $\mathbb{E}[Y^{01} \mid \bar{X}^0, U = 1111] = \mathbb{E}[Y^{01} \mid \bar{X}^0, U = u_1 1 u_3 1]$, $\mathbb{E}[Y^{10} \mid \bar{X}^1, U = 1111] = \mathbb{E}[Y^{10} \mid \bar{X}^1, U = u_1 u_2 1 1]$, *and* $\mathbb{E}[Y^{11} \mid \bar{X}^1, U = 1111] = \mathbb{E}[Y^{11} \mid \bar{X}^1, U = u_1 u_2 u_3 1]$.

*(ii)* $\mathbb{E}[g(X_2^1) \mid X_1, U = 1111] = \mathbb{E}[g(X_2^1) \mid X_1, U = u_1 u_2 u_3 1]$ *and* $\mathbb{E}[g(X_2^0) \mid X_1, U = 1111] = \mathbb{E}[g(X_2^0) \mid X_1, U = u_1 1 u_3 1]$ *for an integrable $g$.*

Monotonicity is a standard assumption which states that for any given patient, their survival status under treatment would be no worse than their survival status had they not received the treatment (Sommer and Zeger, 1991; Follmann, 2006). It is often plausible in studies where providers cannot assign inferior treatments, and it automatically holds when only the fully treated units survive. Assumption 4 implies that the mean of a function of $X_2$ given past data does not depend on future survival. Principal ignorability (Assumption 5) allows us to identify the distribution of outcomes within a principal stratum using an observed stratum, given covariates and treatment assignments; an obvious example of principal ignorability occurs when the characteristics of the always-survivor stratum align with those of the observed survivor population. In our specific case, mean ignorability is sufficient, representing a significantly less stringent condition. Under Assumptions 1, 3, and 4, Assumption 5 simplifies to $\mathbb{E}[Y \mid \bar{X}, \bar{A}, U = 1111] = \mathbb{E}[Y \mid \bar{X}, \bar{A}, S_2 = 1]$ and $\mathbb{E}[g(X_2) \mid X_1, A_1, U = 1111] = \mathbb{E}[g(X_2) \mid X_1, A_1, S_2 = 1]$; the conditional means of always-survivors match those of observed survivors.

## 3 Nonparamteric identification

We define propensity score $e_k^{\bar{a}_k}(\bar{x}_k) = \mathbb{P}(A_k = a_k \mid \bar{x}_k, \bar{a}_{k-1}\bar{C}_{k-1} = \mathbf{0}_{k-1}, \bar{S}_{k-1} = \mathbf{1}_{k-1})$, censoring probability $c_k^{\bar{a}_k}(\bar{x}_k) = \mathbb{P}(C_k = 0 \mid \bar{x}_k, \bar{a}_k, \bar{C}_{k-1} = \mathbf{0}_{k-1}, \bar{S}_{k-1} = \mathbf{1}_{k-1})$, and survival probability $p_k^{\bar{a}_k}(\bar{x}_k) = \mathbb{P}(S_k = 1 \mid \bar{x}_k, \bar{a}_k, \bar{C}_k = \mathbf{0}_k, \bar{S}_{k-1} = \mathbf{1}_{k-1})$ for $k = 1, \ldots, K$. Often, it is more convenient to jointly model $e_k$ and $c_k$. We denote this combined model as $\varphi_k^{\bar{a}_k}(\bar{x}_k) = e_k^{\bar{a}_k}(\bar{x}_k) c_k^{\bar{a}_k}(\bar{x}_k)$. Additionally, we define nuisance models: $\mu_2^{a_1 a_2}(\bar{X}) = \mathbb{E}[Y|\bar{X}, \bar{A} = (a_1, a_2), \bar{C} = (0,0), \bar{S} = (1,1)]$, $m_{p_2}^{a_1 a_2}(X_1) = \mathbb{E}[p_2^{a_1 a_2}(\bar{X})|X_1, A_1 = a_1, C_1 = 0, S_1 = 1]$, $m_{\mu_2}^{a_1 a_2}(X_1) = \mathbb{E}[\mu_2^{a_1 a_2}\mathbf{1}\{A_2 = a_2\}(\bar{X})|X_1, A_1 = a_1, C_1 = 0, S_1 = 1]$. For a fixed policy

$\pi$, the outcome model under $\pi$ is given by $\mu_2^\pi(\bar{X}) = \mu_2^{\pi(\bar{X})}(\bar{X})$. Similarly, we define $e_k^\pi$, $c_k^\pi$, $p_k^\pi$, $m_{p_2}^\pi$, and $m_{\mu_2}^\pi$ for $k = 1, 2$. Table 2 summarizes the notation.

A modified sequential randomization is introduced.

**Assumption 6** (Sequential randomization). *For $k = 1, 2$, (i) $\{S_1^{a_1}, S_2^{a_1 a_2}, C_1^{a_1}, C_2^{a_1 a_2}\} \perp\!\!\!\perp A_k \mid H_k$, (ii) $Y^{a_1 a_2} \perp\!\!\!\perp A_k \mid \{H_k, S_2^{a_1 a_2} = 1\}$, $X_2^{a_1} \perp\!\!\!\perp A_k \mid \{H_k, S_1^{a_1} = 1\}$.*

Assumption 6(i) indicates that counterfactual indicators are independent of treatment assignment, conditional on the history. Assumption 6(ii) implies that, among potential survivors, treatment assignment depends on counterfactual (intermediate) outcomes only through the history. This holds by design in Sequential, Multiple Assignment, Randomized Trials (SMARTs). In observational studies, it holds when all confounders of outcomes and missingness indicators are observed.

In addition, we assume positivity and bounded outcome models.

**Assumption 7.** *For some $\epsilon > 0$ and $L < \infty$, we have, for almost all $\bar{x}$, $\bar{a}$, (i) (Positivity) $e_k^{\bar{a}_k}(\bar{x}), c_k^{\bar{a}_k}(\bar{x}), p_k^{\bar{a}_k}(\bar{x}) \geq \epsilon$, $k = 1, 2$, and (ii) (Bounded mean) $|\mu_2^{a_1 a_2}(\bar{x})|, |m_{\mu_2}^{a_1 a_2}(x_1)| < L$.*

Positivity ensures that propensity scores, censoring and survival probabilities are strictly positive. In practice, both assumptions are generally expected to hold, as typically the outcome of interest, for example, the sequential organ failure score in the MIMIC-III dataset, is inherently bounded.

We further assume that censoring occurs according to the missing at random (MAR) mechanism.

**Assumption 8.** *For $k = 1, 2$ and $a_1, a_2, a_1', a_2' \in \{0, 1\}$, (i) $S_2^{a_1 a_2} \perp\!\!\!\perp C_2^{a_1' a_2'} \mid H_2$, $\bar{S}^{a_1 a_2} \perp\!\!\!\perp \bar{C}^{a_1' a_2'} \mid X_1$, (ii) $Y^{a_1 a_2} \perp\!\!\!\perp \bar{C}_k^{\bar{a}_k} \mid \{H_k^{\bar{a}_k}, S_k^{\bar{a}_k} = 1\}$, $X_2^{a_1} \perp\!\!\!\perp C_1^{a_1} \mid \{X_1, S_1^{a_1} = 1\}$.*

Assumption 8 states that, conditional on the history, censoring is non-informative when estimating survival probabilities, and when estimating outcomes among potential survivors. With the addition of Assumption 8, Assumption 5 simplifies to $\mathbb{E}[Y \mid \bar{X}, \bar{A}, U = 1111] = \mathbb{E}[Y \mid \bar{X}, \bar{A}, C_2 = 0, S_2 = 1]$ and $\mathbb{E}[g(X_2) \mid X_1, A_1, U = 1111] = \mathbb{E}[g(X_2) \mid X_1, A_1, C_2 = 0, S_2 = 1]$, respectively, thereby identifying the always-survivor distributions of outcomes via observed data.

In the first theorem, we identify the always-survivor value function with no restriction imposed on the distribution of the variables other than principal ignorability.

**Theorem 1.** *For a fixed policy $\pi = (\pi_1, \pi_2)$, under assumptions 1 and 3-8, the always-survivor value is identified as*

$$V_{\text{AS}}(\pi) = \frac{\mathbb{E}[p_1^0(X_1) m_{p_2}^{00}(X_1) m_{\mu_2}^\pi(X_1)]}{\mathbb{E}[p_1^0(X_1) m_{p_2}^{00}(X_1)]} \tag{1}$$

$$= \mathbb{E}\left[p_1^0(X_1) m_{p_2}^{00}(X_1) \frac{\mathbf{1}\{\bar{A} = \pi(\bar{X})\}(1 - C_1)(1 - C_2) S_1 S_2}{\varphi_1^\pi(X_1) \varphi_2^\pi(\bar{X}) p_1^\pi(X_1) p_2^\pi(\bar{X})} Y\right] \Big/ \mathbb{E}[p_1^0(X_1) m_{p_2}^{00}(X_1)]. \tag{2}$$

Identification (1) expresses the always-survivor value as a weighted mean of the outcome model for observed survivors, where the weight is given by the principal score $\omega(X_1) = \Pr(U = 1111 \mid X_1)/\Pr(U = 1111) = p_1^0(X_1) m_{p_2}^{00}(X_1)/\mathbb{E}[p_1^0(X_1) m_{p_2}^{00}(X_1)]$. If all trajectories are always-survivors and no censoring occurs, then $\omega(X_1) = 1$, and (1) reduces to the complete-case Q-learning, or outcome regression (OR), methodology. Similarly, identification (2) is analogous to the inverse probability weighting (IPW) identification when all units are always-survivors.

A straightforward estimator based on this observation is *principal Q-learning*, which models the always-survivor value using survivor Q-functions and the principal score. Consider the estimator $\widehat{V}_Q(\pi) = \mathbb{E}_n[\widehat{\omega}(X_1) \widehat{m}_{\mu_2}^\pi(X_1)]$, where $\mathbb{E}_n$ denotes the empirical mean. As is typical in Q-learning, the conditional outcome $\widehat{m}_{\mu_2}^\pi(X_1)$ can be fitted backward-recursively:

$$Q_2(\bar{x}; \pi) \equiv \hat{\mu}_2^\pi(\bar{x}) = \widehat{\mathbb{E}}\left[Y \mid \bar{x}, \pi(\bar{x}), \bar{C} = \mathbf{0}_2, \bar{S} = \mathbf{1}_2\right], \quad \tilde{V}^\pi := Q_2(\bar{x}; \pi),$$

$$Q_1(x_1; \pi) \equiv \widehat{m}_{\mu_2}^\pi(X_1) = \widehat{\mathbb{E}}\left[\tilde{V}^\pi \mid x_1, \pi_1(x_1), C_1 = 0, S_1 = 1\right].$$

Principal Q-learning requires correct specification of the outcome models to ensure consistency with the always-survivor value function. In the following section, we aim to derive an estimator based on the efficient influence function, which can offer protection against misspecification of these outcome models.

Table 3: Five scenarios ensuring the consistency of (3). 'X' indicates correct specification.

| | $\widehat{\varphi}_1$ | $\widehat{p}_1$ | $\widehat{\varphi}_2$ | $\widehat{p}_2$ | $\widehat{m}_{p_2}$ | $\widehat{\mu}_2$ | $\widehat{m}_{\mu_2}$ |
|------|---|---|---|---|---|---|---|
| (i)   | X | X | X | X | X |   |   |
| (ii)  | X |   | X |   |   | X | X |
| (iii) | X | X |   | X | X | X |   |
| (iv)  | X |   |   | X |   | X | X |
| (v)   |   | X |   | X | X | X | X |

## 4   Multiply robust off-policy evaluation

Starting from the identification formula, we derive the efficient influence function (EIF) of $V_{\text{AS}}(\pi)$, from which the semiparametric efficiency bound follows. Let $A_0 = C_0 = 0$ and $\varphi_0(\cdot) = 1$. Define $D(O) = \mathbb{E}[p_1^0(X_1)m_{p_2}^{00}(X_1)]$ and $N(O;\pi) = \mathbb{E}[p_1^0(X_1)m_{p_2}^{00}(X_1)m_{\mu_2}^{\pi}(X_1)]$ be the denominator and the numerator of identification (1) given the observed data $O$ and a polity $\pi$, respectively. Theorem 2 characterizes the EIF of (1).

**Theorem 2.** *Under assumptions 1 and 3-8, the efficient influence function of $V_{AS}(\pi)$ is $\psi_{V(\pi)}(O) = \left\{ \phi_{N(\pi)}(O) - V_{AS}(\pi)\phi_D(O) \right\}/D(O)$ where*

$$\phi_D(O) = \frac{\mathbf{1}\{\bar{A} = \bar{C} = \mathbf{0}_2\}}{\varphi_1^0(X_1)\varphi_2^{\mathbf{0}_2}(\bar{X})}S_1 S_2 + \sum_{k=1}^{2}\left\{ \prod_{j=0}^{k-1}\frac{\mathbf{1}\{A_j = C_j = 0\}}{\varphi_j^{\mathbf{0}_j}(\bar{X}_j)} - \prod_{j=0}^{k}\frac{\mathbf{1}\{A_j = C_j = 0\}}{\varphi_j^{\mathbf{0}_j}(\bar{X}_j)} \right\}Q_{S,k},$$

$$\phi_{N(\pi)}(O) = Q_{Y,1}\phi_D(O) + \sum_{k=1}^{2}\prod_{j=1}^{k}\frac{\mathbf{1}\{\pi_j(\bar{X}_j) = A_j\}(1 - C_j)S_j}{\varphi_j^{\pi}(\bar{X}_j)p_j^{\pi}(\bar{X}_j)}\left(Q_{Y,j+1} - Q_{Y,j}\right)Q_{S,1},$$

*with $Q_{S,j}$, $Q_{Y,j}$ defined as in the Appendix A.1.2. Thus, the semiparametric efficiency bound for $V_{AS}(\pi)$ is $\Upsilon(\pi) = \mathbb{E}\left[\psi_{V(\pi)}^2(O)\right]$.*

Based on the EIF, we construct a multiply robust (MR) estimator for $V_{\text{AS}}(\pi)$:

$$\widehat{V}_{\text{MR}}(\pi) = \frac{\mathbb{E}_n\{\hat{\phi}_{N(\pi)}(O)\}}{\mathbb{E}_n\{\hat{\phi}_D(O)\}} \tag{3}$$

where $\hat{\phi}_N$ and $\hat{\phi}_D$ are plug-in estimators derived from the estimated nuisance models $\hat{p}_k^{\bar{a}_k}$, $\hat{p}_k^{\bar{a}_k}$, $\hat{\mu}_2^{a_1 a_2}$, $\hat{m}_{p_2}^{a_1 a_2}$, and $\hat{m}_{\mu_2}^{a_1 a_2}$ for $k = 1, 2$. It can be observed that if all observations are always-survivors and there is no censoring, the MR estimator simplifies to the standard augmented inverse probability weighting (AIPW) estimator. Furthermore, the estimator reduces to the estimators proposed by Jiang et al. (2022) and Chu et al. (2023) in single decision point cases.

Let $e_k^{\bar{a}_k}(\cdot; \hat{\alpha}_k)$, $c_k^{\bar{a}_k}(\cdot; \hat{\eta}_k)$, $p_k^{\bar{a}_k}(\cdot; \hat{\gamma}_k)$, $\mu_2^{\bar{a}}(\cdot; \hat{\zeta})$, $m_{p_2}^{\bar{a}}(\cdot; \hat{\xi})$, $m_{\mu_2, \pi}^{\bar{a}}(\cdot; \hat{\nu})$ for $k = 1, 2$ denote the parametric models for the corresponding nuisance models and let $\alpha_k^*, \eta_k^*, \gamma_k^*, \zeta^*, \xi^*, \nu^*$ be the limit of their parameters. Let $\widehat{\phi}_N, \widehat{\phi}_D$ be the plug-in estimator of $\phi_N, \phi_D$, respectively. Assume the uniform weak law of large numbers holds, i.e., $\mathbb{E}_n(\hat{\phi}_D) \xrightarrow{\mathbb{P}} \mathbb{E}(\phi_D)$ and $\mathbb{E}_n(\hat{\phi}_{N(\pi)}) \xrightarrow{\mathbb{P}} \mathbb{E}(\phi_{N(\pi)})$.

**Theorem 3.** *Under Assumptions 1, 3-8, and the regularity conditions described in the Appendix, $\widehat{V}_{MR}(\pi) = \mathbb{E}_n\{\hat{\phi}_{N(\pi)}(O)\}/\mathbb{E}_n\{\hat{\phi}_D(O)\}$ is multiply robust, meaning $\widehat{V}_{MR}(\pi) \xrightarrow{p} V_{AS}(\pi)$ as $n \to \infty$ if any of the scenarios in Table 3 are satisfied. Furthermore, $\widehat{V}_{MR}(\pi)$ achieves the semiparametric efficiency bound when all nuisance models are correctly specified.*

Theorem 3 highlights the dual benefits of the proposed estimator: robustness to partial model misspecification and local efficiency under correct specification. We conclude this section by presenting that the estimator exhibits multiple robustness and achieves semiparametric local efficiency with flexible nonparametric modeling. Let $\hat{\theta} = \{\widehat{\varphi}_k^{\bar{a}_k}, \widehat{p}_k^{\bar{a}_k}, \widehat{\mu}_2^{a_1 a_2}, \widehat{m}_{p_2}^{00}, \widehat{m}_{\mu_2}^{a_1 a_2} : \bar{a} \in \{0,1\}^2\}$ be the collection of nonparametric models, and let $\theta$ represent the true data-generating process.

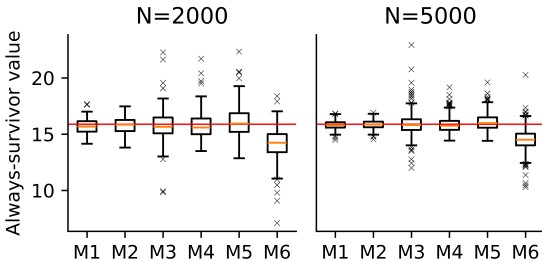

Figure 1: Value estimates under a fixed policy. The red line is drawn at the true value.

Table 4: Off-policy evaluation result. S.E. indicates standard error.

|  |  | M1 | M2 | M3 | M4 | M5 | M6 |
|---|---|---|---|---|---|---|---|
| **n=2000** | **Bias** | -0.20 | -0.10 | -0.15 | -0.13 | 0.22 | -1.74 |
|  | **S.E.** | 0.63 | 0.63 | 1.19 | 1.11 | 1.26 | 1.27 |
| **n=5000** | **Bias** | -0.07 | -0.02 | -0.04 | -0.06 | 0.18 | -1.39 |
|  | **S.E.** | 0.37 | 0.37 | 1.05 | 0.66 | 0.74 | 1.04 |

**Theorem 4.** *Assume that the nuisance models are fitted using sample-splitting or that the components in $\hat{\theta}$ belong to a Donsker class. Under Assumptions 1, 3-8, and the regularity conditions described in the Appendix, $n^{1/2}\big(\widehat{V}_{MR}(\pi) - V_{AS}(\pi)\big) \xrightarrow{d} \mathcal{N}\left(0, \Upsilon(\pi)\right)$*

The results provide a foundation for leveraging flexible machine learning methods to accurately model the nuisance components in practice. Furthermore, sample-splitting or cross-fitting (Chernozhukov et al., 2018) provides a practical technique for guaranteeing the property in Theorem 4. A cross-fitting algorithm is detailed in the Appendix.

## 5   Multiply rubust off-policy learning

Building upon the theoretical guarantees, we propose a method to learn an always-survivor-optimal policy in this section. While a variety of policy classes can be accommodated, we focus our analysis on the class of linear policies, where $\pi_\beta = \{\pi_{\beta_1}, \pi_{\beta_2}\}$ is determined by a linear combination of observed characteristics, specifically $\pi_{\beta_k} = \mathbf{1}\{\tilde{h}_k^\top \beta_k > 0\}$, where $\tilde{h}_k^\top = (1, h_k^\top)$. This approach is widely used in practice, as the coefficients $\beta = (\beta_1, \beta_2)$ provide a clear understanding of how covariates influence treatment decisions. To simplify notation, we define $V_{AS}(\beta) \equiv V_{AS}(\pi_\beta)$ as the always-survivor value under $\pi_\beta$. Let $\hat{\beta}$ be the parameter that maximizes $\widehat{V}_{MR}(\beta)$, and let $\beta^*$ be the probability limit of $\hat{\beta}$.

**Theorem 5.** *Assume that the nuisance models are fitted using sample-splitting, or that the components in $\hat{\theta}$ belong to a Donsker class. Under Assumptions 1, 3-8, and the regularity conditions described in the Appendix, $n^{1/2}\big(\widehat{V}_{MR}(\hat{\beta}) - V_{AS}(\beta^*)\big) \xrightarrow{d} \mathcal{N}\left(0, \Upsilon(\beta^*)\right)$.*

Therefore, the MR estimator yields a policy with semiparametric local efficiency, guaranteeing optimal asymptotic variance when the models are correctly specified. Even with partial misspecification, the framework inherits multiple robustness, allowing decision-makers to confidently search for optimal treatment regimes, providing increased protection against model misspecification. This is particularly valuable in real-world applications where the true underlying models may be complex or unknown. Additionally, a cross-fitting estimator can be employed to ensure the results presented in Theorem 5.

## 6   Simulation study

To evaluate the performance of the proposed methodology, we simulated trajectories with monotone censoring and truncation by death, using varying rates of missingness. We generated $X_1$ from a

continuous uniform distribution over the interval $[-0.3, 0.7]$. $A_k$, $C_k^{\bar{a}_k}$, and $S_k^{\bar{a}_k}$ were generated using logistic models and the intermediate $X_2^{a_1}$ and final outcome $Y^{a_1 a_2}$ were generated from normal distributions. This setup resulted in censoring rates ranging from 8% to 13% and survival rates between 65% and 70%. We conducted experiments with sample sizes of $n = 2000$ and $n = 5000$.

**Multiple robustness** To demonstrate the multiple robustness, we conducted experiments across five model specification scenarios (M1-M5) expected to yield consistency, and one scenario (M6) expected to fail. Each process was repeated 500 times, and $\widehat{V}_{\text{MR}}$ was computed in all six model specification scenarios. True always-survivor value were computed by plugin estimator with known true models. The Appendix provides detailed information on data generation and model specification scenarios, including results from additional experiments.

Figure 1 and Table 4 presents the results. Across scenarios M1-M5, the MR estimator consistently yields estimates close to the true value, demonstrating its multiple robustness. Based on the EIF, 95% confidence intervals for the always-survivor value in scenario M1 were computed analytically. The coverage rates, 94.2% for $n = 2000$ and 95.2% for $n = 5000$, are close the nominal value.

The same simulation was also run with estimators based on the OR (1) and the IPW (2) identifications. The standard error for the MR estimator (0.631 for $N = 2000$, 0.369 for $N = 5000$) was positioned between the OR estimator (0.475 for $N = 2000$, 0.280 for $N = 5000$) and the IPW estimator (2.250 for $N = 2000$, 1.346 for $N = 5000$). Our results confirm that the MR estimator achieves comparable efficiency to the OR estimator.

**Off-policy learning** In each of 500 independent data replications, we determined the linear policy $\hat{\beta}_{\text{MR}}$ and $\hat{\beta}_{\text{AIPW}}$ by maximizing $\widehat{V}_{\text{MR}}(\beta)$ or $\widehat{V}_{\text{AIPW}}(\beta)$ respectively, where

$$
\begin{aligned}
\widehat{V}_{\text{AIPW}}(\pi) = \mathbb{E}_n \Big\{ & \frac{\mathbf{1}\{\bar{A} = \pi(\bar{X})\}(1 - \tilde{C}_1)(1 - \tilde{C}_2)}{\tilde{\varphi}_1^\pi(X_1)\tilde{\varphi}_2^\pi(\bar{X})} \left\{ Y - Q_2(\bar{X}, \bar{A}) \right\} \\
& + \frac{\mathbf{1}\{A_1 = \pi_1(X_1)\}(1 - \tilde{C}_1)}{\tilde{\varphi}_1^\pi(X_1)} \left\{ Q_2(\bar{X}, \bar{A}) - Q_1(X_1, A_1) \right\} + Q_1(X_1, A_1) \Big\},
\end{aligned}
\tag{4}
$$

$\tilde{C}_k = C_k(1 - S_k)$, $\tilde{\varphi}_k^{\bar{a}_k}(\bar{x}_k) = \widehat{\mathbb{P}}(\tilde{C}_k = 0|\bar{x}_k, \bar{a}_k, \tilde{C}_l = 0, l = 1, \ldots, k - 1)$, and $Q_k(\bar{x}_k, \bar{a}_k) = \widehat{\mathbb{E}}(Y|\bar{x}_k, \bar{a}_k, \tilde{C}_l = 0, l = 1, \ldots, k)$. This is a standard approach that treats trajectories experiencing death as censored observations.

We employed a differential evolution algorithm to perform the value search. The learned policies are compared with the true optimal linear policy $\beta^*$, obtained by maximizing the plug-in estimator of (1) using a large dataset under ground truth nuisance models. Figure 2 compares the estimated values $\widehat{V}_{\text{MR}}(\hat{\beta}_{\text{MR}})$ and $\widehat{V}_{\text{AIPW}}(\hat{\beta}_{\text{AIPW}})$, the values $V(\hat{\beta}_{\text{MR}})$ and $V(\hat{\beta}_{\text{AIPW}})$ evaluated with the learned policies, and the true optimal value $V(\beta^*)$. The 95% confidence intervals based on the EIF exhibit coverage rates of 94.8% for $n = 2000$ and 96% for $n = 5000$, close to the nominal value.

The percentage of correct decisions on the always-survivors (PCD-AS) was calculated using an independently generated large dataset. The detailed calculation is described in the Appendix. PCD-AS shows a convergence trend towards one as the training set size increases. The average PCD-AS of the MR estimator was 0.992 for $n = 2000$ and 0.994 for $n = 5000$, with standard deviations of 0.007 and 0.004, respectively, which is closer to one with less variability than the PCD-AS of the AIPW estimator, averaging 0.987 and 0.988 with standard deviations of 0.008 and 0.007, respectively.

# 7 Analysis of MIMIC-III data

To illustrate the utility of our proposed methodology, we applied it to the Medical Information Mart for Intensive Care III (MIMIC-III) v1.4 database. MIMIC-III is a publicly accessible, MIT-licensed database containing de-identified health records from over 40,000 patients admitted to critical care units at Beth Israel Deaconess Medical Center between 2001 and 2012. This dataset encompasses comprehensive patient information, including vital signs, laboratory results, medications, and survival outcomes, making it a suitable resource. Johnson et al. (2016) provides a detailed description.

To simplify the analysis, we focused on two time points post-sepsis onset. We utilized eight baseline variables - age, weight, temperature (°C), log glucose, log blood urea nitrogen (BUN), log creatinine (mg/dL), log white blood cell count (WBC), and log sequential organ failure assessment

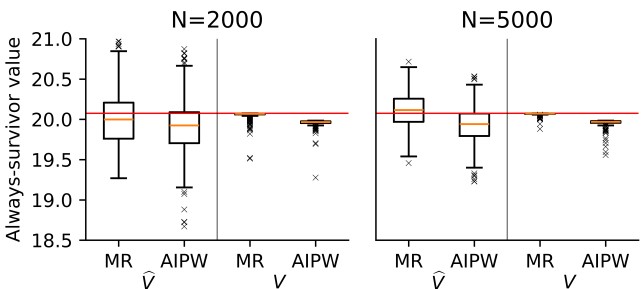

Figure 2: Value estimates from 500 independent simulation runs. Left panels of each plot show the estimated values ($\widehat{V}_{\text{MR}}(\hat{\beta}_{\text{MR}})$ and $\widehat{V}_{\text{AIPW}}(\hat{\beta}_{\text{AIPW}})$), while right panels show the corresponding values evaluated with true nuisance functions ($V(\hat{\beta}_{\text{MR}})$ and $V(\hat{\beta}_{\text{AIPW}})$). The red horizontal line indicates the true optimal value ($V(\beta^*)$).

(SOFA) score - as $X_1$. For $X_2$, we used seven intermediate variables, which were the same as the baseline covariates excluding age. Intervention with mechanical ventilation at each time point was represented by $A_1$ and $A_2$. The negative SOFA score at the final time point was used as the outcome $Y$, so that a higher value indicates a better condition. If a patient died within $k$th 24-hour frame, $S_k = 0$; otherwise, $S_k = 1$. If a patient did not die but the outcome is missing, at the follow-up point, $C_k = 1$. We focused on the high-risk group by using patient data with a baseline SOFA score above 8. Processing procedure is detailed in the Appendix.

We employed logistic regression models for estimating the propensity score, censoring and survival probability. For continuous outcome models, we fitted random forest regressors. Lastly, generalized additive models were fitted to estimate the conditional mean functions, $m_{p_2}$ and $m_{\mu_2}$. We used a differential evolution algorithm to optimize within the class of linear policies.

To evaluate the benefit of our proposed estimator, we conducted a repeated 50 iterations of train-test split. In each iteration, stratified sampling on censoring ($C_1, C_2$) and survival ($S_1, S_2$) indicators was used to create balanced training and test sets. Policies were learned on training data and their value estimated on test data. We established the "true" optimal value on the test set with multiply robust estimator as a benchmark. Figure 3 summarizes the results.

The MR estimator obtains policy closer to the true optimum than the AIPW estimator. Consistent with recent findings (Sarraf et al., 2024), the learned policy (Figure 3, right panel) demonstrates a significant influence of age and weight on treatment decisions. Specifically, patients with higher age and weight were more likely to be assigned to active treatment at baseline. At the intermediate stage, the learned treatment regime predominantly assigned patients to the untreated group. This observation suggests the acute nature of sepsis necessitates rapid intervention, while also indicating a cautious approach to mechanical ventilation, likely due to its associated risks (Unroe et al., 2010).

Finally, we note that despite its role as a benchmark, the clinical utility of sepsis policies derived from MIMIC-III data is actively debated. Nauka et al. (2025) found applying trained models in practice inadequate, stating that missing data and diagnostic uncertainty lead to unpredictable model behavior. Conversely, Festor et al. (2022) claimed the model recommended fewer hazardous decisions than human clinicians despite data missingness.

## 8 Discussions

**Indentification for the multiple decision points case**   Consider a scenario with $K \geq 3$ decision points. The identification formula for the always-survivor value, with corresponding generalized assumptions, is given by

$$V_{\text{AS},K}(\pi) = \mathbb{E}[\prod_{k=1}^{K} m_{p_k}^{\mathbf{0}_k}(X_1) m_{\mu_K}^{\pi}(X_1)] / \mathbb{E}[\prod_{k=1}^{K} m_{p_k}^{\mathbf{0}_k}(X_1)] =: N_K(\pi)/D_K, \qquad (5)$$

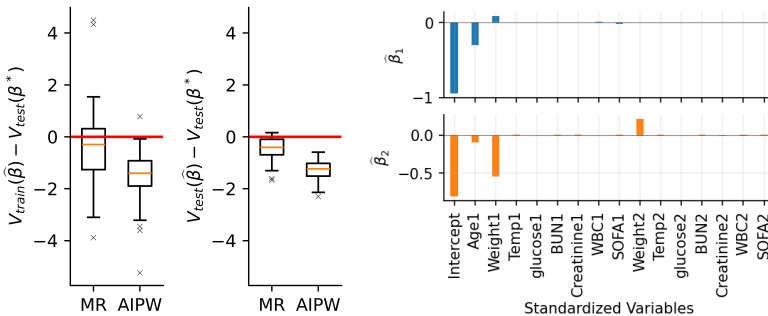

Figure 3: Analysis of MIMIC-III database. (Left) Training set value was evaluated by nuisance model learned on training data, while test set value was computed by nuisance models learned on test data. Both values were evaluated on the test set. The left panel displays the discrepancy between the training set value function with training set optimal policy ($\hat{\beta}$) and the test set value function with test set optimal policy ($\beta^*$). The right panel compares the test set value function achieved by $\hat{\beta}$ and $\beta^*$. A solid red line is drawn at zero. (Right) Policy learned from the MR estimator.

where, for $k = 1, \ldots, K$, the nuisance models are defined as $p_k^{\bar{a}_k}(\bar{x}_k) := \mathbb{P}(S_k | \bar{x}_k, \bar{a}_k, \bar{C}_k = \mathbf{0}_k, \bar{S}_{k-1} = \mathbf{1}_{k-1})$, $\mu_K^{\bar{a}}(\bar{x}) := \mathbb{E}[Y | \bar{x}, \bar{a}, \bar{C} = \mathbf{0}_K, \bar{S} = \mathbf{1}_K]$, $m_{p_k}^{\mathbf{0}_k}(x_1) := \mathbb{E}[p_k^{\mathbf{0}_k}(\bar{X}_k) | x_1, 0, C_1 = 0, S_1 = 1]$, and $m_{\mu_K}^{\pi}(x_1) := \mathbb{E}[\mu_K^{\pi}(\bar{X}_K) \mathbf{1}\{\bar{A} = \pi(\bar{X}_K)\} | x_1, \pi_1, C_1 = 0, S_1 = 1]$. The outcome model can again be estimated backward-recursively. The EIF derivation follows a process similar to Theorem 2, albeit with significant effort. Due to its technical complexity, the EIF is presented without proof. Define $Q_{S,j}$ and $Q_{Y,j}$ as in the Appendix A.1.2. Let

$$\phi_{D_K}(O) = \prod_{k=1}^{K} \frac{\mathbf{1}\{A_k = C_k = 0\}}{\varphi_k^{\mathbf{0}_k}(\bar{X}_k)} S_k + \sum_{k=1}^{K} \left\{ \frac{\mathbf{1}\{\bar{A}_{k-1} = \bar{C}_{k-1} = \mathbf{0}_{k-1}\}}{\prod_{j=0}^{k-1} \varphi_j^{\mathbf{0}_j}(\bar{X}_j)} - \frac{\mathbf{1}\{\bar{A}_k = \bar{C}_k = \mathbf{0}_k\}}{\prod_{j=0}^{k} \varphi_j^{\mathbf{0}_j}(\bar{X}_j)} \right\} Q_{S,k},$$

$$\phi_{N_K(\pi)}(O) = Q_{Y,1} \phi_{D_K}(O) + \sum_{k=1}^{K} \prod_{j=1}^{k} \frac{\mathbf{1}\{\pi_j(\bar{X}_j) = A_j\}(1 - C_j) S_j}{\varphi_j^{\pi}(\bar{X}_j) p_j^{\pi}(\bar{X}_j)} (Q_{Y,j+1} - Q_{Y,j}) Q_{S,1}.$$

The EIF of (5) is given as $\{\phi_{N_K(\pi)}(O) - V_{\mathrm{AS},K}(\pi) \phi_{D_K}(O)\} / D_K(O)$.

Verifying the assumptions and accurately fitting the nuisance models become challenging with a growing number of decision points. The pessimism principle (Jin et al., 2021) or a minimax learning approach (Kallus and Zhou, 2021) could potentially offer a solution, warranting future investigation.

**Practical considerations** If stratum assignment were known, a policy $\pi$ maximizing (3) could directly optimize average outcomes within always-survivors. For protectables, active treatment would be optimal, as survival is contingent on it. Since $U$ is latent, one approach is to estimate the always-surviving probability based on patient characteristics using the result $\mathbb{P}(U = 1111 | x_1) = p_1^0(x_1) m_{p_2}^{00}(x_1)$. Similarly, let $T^{\bar{a}}$ denote the potential survival time had a subject received treatment $\bar{a}$. Following Jiang et al. (2017), we can derive an estimator for $S_t^{*\bar{a}}(X_1) = \mathbb{E}[\mathbb{P}(T^{\bar{a}} > t | \bar{X}) | X_1]$. By evaluating $\widehat{S}_{t_0}^{*\mathbf{0}_K}(x_1)$ at a time $t_0$ beyond the final decision point, we estimate the probability of a subject being an always-survivor until $Y$ is observed. $\pi$ is then applied to patients with $\hat{p}_1^0(x_1) \hat{m}_{p_2}^{00}(x_1) > c$ or $\widehat{S}_{t_0}^{*\mathbf{0}_K}(x_1) > c$, where $c$ is a predefined threshold (e.g., 0.95).

Alternatively, we could balance between survivor-optimal decisions and overall population survival probability, ensuring policy's applicability to all subjects. We consider the penalized problem: $\tilde{\pi} = \mathrm{argmax}_{\pi \in \Pi} \widehat{V}_{\mathrm{MR}}(\pi) + \lambda(1 - \widehat{S}_t^*(\pi))$ where $S_t^*(\pi) = \mathbb{E}[\mathbb{P}(T^{\pi} > t | \bar{X})]$. Intuitively, the potential $t$-year mortality rate under $\pi$ acts as a regularizing penalty. The problem is well described in Zhou et al. (2021). $\tilde{\pi}$ is not survivor-optimal but controls the survival rate to a level determined by $\lambda$.

Finally, one could explore a principal stratification framework based on potential time to death. Such an approach would allow for more nuanced and fine-grained classification of participants beyond the conventional binary survival status, though it introduces additional methodological and inferential challenges (Zhang and Yang, 2025).

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

# A  Appendix

Section A.1 presents the regularity conditions and the proofs of the theorems stated in the main text.

Section A.2 provides technical details related to the cross-fitting algorithm, the experimental settings for the simulation study, and the preprocessing procedure of the MIMIC-III data. Results from an additional experiment are provided.

## A.1  Proof of theoretical results

### A.1.1  Proof of Theorem 1

Throughout the appendix, for a given policy $\pi$, we simplify the notation by writing $N = N(\pi)$, $V(\pi) = V_{\mathrm{AS}}(\pi)$, and $\widehat{V}(\pi) = \widehat{V}_{\mathrm{MR}}(\pi)$. The first lemma demonstrates that the conditional mean of a quantity within the always-survivor stratum is equal to the unconditional mean of the function multiplied by the principal score.

**Lemma 1.** *For a function $g(X_1)$ such that $\mathbb{E}|g(X_1)| < \infty$, we have*

$$\mathbb{E}\left[g(X_1) \mid U = 1111\right] = \mathbb{E}\left[\frac{p_1^0(X_1)m_{p_2}^{00}(X_1)}{\mathbb{E}\left[p_1^0(X_1)m_{p_2}^{00}(X_1)\right]}g(X_1)\right].$$

*Proof of Lemma 1.* First, by Bayes' theorem,

$$\mathbb{E}[g(X_1) \mid U = 1111] = \mathbb{E}\left[\frac{f(X_1 \mid U = 1111)}{f(X_1)}g(X_1)\right] = \mathbb{E}\left[\frac{\mathbb{P}(U = 1111 \mid X_1)}{\mathbb{P}(U = 1111)}g(X_1)\right].$$

In addition,

$$
\begin{aligned}
\mathbb{P}(U = 1111|X_1) &= \mathbb{P}(S_2^{00} = 1|X_1) \\
&= \mathbb{P}(S_2^{00} = 1|X_1, A_1 = C_1 = 0) && \text{(Assumptions 6,8(i))} \\
&= \mathbb{E}\left[\mathbb{P}(S_2^{00} = 1|X_1, X_2^0, \bar{A} = (0,0), \bar{C}^{00} = (0,0)) \mid X_1, A_1 = C_1 = 0\right] && \text{(Assumptions 6,8(ii))} \\
&= \mathbb{E}\left[\mathbb{P}(S_2 = 1|\bar{X}, \bar{A} = \bar{C} = (0,0)) \mid X_1, A_1 = C_1 = 0\right] && \text{(Assumption 1)} \\
&= \mathbb{E}\left[\mathbb{P}(S_1 = S_2 = 1|\bar{X}, \bar{A} = \bar{C} = (0,0)) \mid X_1, A_1 = C_1 = 0\right] && \text{(Assumption 3)} \\
&= \mathbb{E}\left[\mathbb{P}(S_2 = 1|\bar{X}, \bar{A} = \bar{C} = (0,0), S_1 = 1)\mathbb{P}(S_1 = 1|\bar{X}, \bar{A} = \bar{C} = (0,0)) \mid X_1, A_1 = C_1 = 0\right] \\
&= \mathbb{E}\left[\mathbb{P}(S_2 = 1|\bar{X}, \bar{A} = \bar{C} = (0,0), S_1 = 1)\mathbb{P}(S_1 = 1|\bar{X}, A_1 = C_1 = 0) \mid X_1, A_1 = C_1 = 0\right] \\
& && \text{(Assumptions 6,8(ii))} \\
&= \mathbb{E}\bigg[\mathbb{P}(S_2 = 1|\bar{X}, \bar{A} = \bar{C} = (0,0), S_1 = 1) \\
&\qquad \times \mathbb{P}(S_1 = 1|X_1, A_1 = C_1 = 0)\frac{f(X_2|X_1, A_1 = C_1 = 0, S_1 = 1)}{f(X_2|X_1, A_1 = C_1 = 0)}\bigg|X_1, A_1 = C_1 = 0\bigg] \\
&= \mathbb{E}\big[\mathbb{P}(S_2 = 1|\bar{X}, \bar{A} = \bar{C} = (0,0), S_1 = 1) \\
&\qquad \times \mathbb{P}(S_1 = 1|X_1, A_1 = C_1 = 0)|X_1, A_1 = C_1 = 0, S_1 = 1\big] \\
&= p_1^0(X_1)\mathbb{E}\left[p_2^{00}(\bar{X}) \mid X_1, A_1 = C_1 = 0, S_1 = 1\right] \\
&= p_1^0(X_1)m_{p_2}^{00}(X_1),
\end{aligned}
$$

and similarly, $\mathbb{P}(U = 1111) = \mathbb{E}\left[p_1^0(X_1)m_{p_2}^{00}(X_1)\right]$. $\qquad\square$

Theorem 1 follows directly from Lemma 1.

*Proof of Theorem 1.*

$$\begin{aligned}
V(\pi) &= \mathbb{E}[Y^\pi | U = 1111] \\
&= \mathbb{E}\left[\mathbb{E}(Y^\pi | X_1, U = 1111) \mid U = 1111\right] \\
&= \mathbb{E}\left[\mathbb{E}(Y^\pi | X_1, A_1 = \pi_1(X_1), C_1 = 0, S_1 = 1, U = 1111) \mid U = 1111\right] \\
&\qquad\qquad\qquad\qquad\qquad\qquad\qquad\qquad\qquad\text{(Assumptions 3,6,8)} \\
&= \mathbb{E}\left[\mathbb{E}\left[\mathbb{E}(Y^\pi | \bar{X}, A_1 = \pi_1(X_1), C_1 = 0, S_1 = 1, U = 1111) \mid X_1, U = 1111\right] \mid U = 1111\right] \\
&= \mathbb{E}\Bigg[\mathbb{E}\big[\mathbb{E}(Y^\pi | \bar{X}, \bar{A} = \pi(\bar{X}), \bar{C} = \bar{0}, \bar{S} = \bar{1}, U = 1111) \\
&\qquad\qquad \big| X_1, A_1 = \pi_1(X_1), C_1 = 0, S_1 = 1, U = 1111\big] \Big| U = 1111\Bigg] \\
&\qquad\qquad\qquad\qquad\qquad\qquad\qquad\qquad\qquad\text{(Assumptions 3,6,8)} \\
&= \mathbb{E}\left[\mathbb{E}\left[\mu_2^\pi(\bar{X}) \mid X_1, A_1 = \pi_1(X_1), C_1 = 0, S_1 = 1\right] \mid U = 1111\right] \qquad \text{(Assumption 5)} \\
&= \mathbb{E}\left[m_{\mu_2}^\pi(X_1) \mid U = 1111\right]. \qquad\qquad\qquad\qquad \text{(Assumptions 3,4,5,6,8)}
\end{aligned}$$

Alternatively,

$$\begin{aligned}
V(\pi) &= \mathbb{E}\Bigg[\mathbb{E}\big[\mathbb{E}(Y^\pi | \bar{X}, \bar{A} = \pi(\bar{X}), \bar{C} = \bar{0}, \bar{S} = \bar{1}, U = 1111) \\
&\qquad\qquad \big| X_1, A_1 = \pi_1(X_1), C_1 = 0, S_1 = 1, U = 1111\big] \Big| U = 1111\Bigg] \\
&= \mathbb{E}\left[\mathbb{E}\left\{\frac{\mathbf{1}\{\bar{A} = \pi(\bar{X})\}(1 - C_1)(1 - C_2)S_1 S_2}{\varphi_1^\pi(X_1)\varphi_2^\pi(\bar{X})p_1^\pi(X_1)p_2^\pi(\bar{X})} Y \Big| X_1\right\}\right]. \qquad \text{(Assumption 5)}
\end{aligned}$$

Hence, the result follows by Lemma 1. $\qquad\qquad\qquad\qquad\qquad\qquad\qquad\qquad\square$

### A.1.2  Proof of theorem 2

First, we define $Q_{S,j}$ and $Q_{Y,j}$ in Theorem 2 as

$$\begin{aligned}
Q_{S,1} &= \mathbb{E}[S_1 p_2^{00}(\bar{X})|X_1, A_1 = 0, C_1 = 0] = p_1^0(X_1)m_{p_2}^{00}(X_1), \quad Q_{S,2} = S_1 p_2^{00}(\bar{X}), \\
Q_{Y,1} &= m_{\mu_2}^\pi(X_1), \quad Q_{Y,2} = \mu_2^\pi(\bar{X}), \quad Q_{Y,3} = Y.
\end{aligned}$$

For $K \geq 3$ cases, we similarly define

$$\begin{aligned}
Q_{S,K} &= \prod_{j=1}^{K-1} S_j p_K^{\mathbf{0}_K}(\bar{X}), \quad Q_{S,j} = \mathbb{E}[Q_{S,j+1}|\bar{X}_j, \bar{A}_j = \mathbf{0}_j, \bar{C}_j = \mathbf{0}_j], \\
Q_{Y,K+1} &= Y, \quad Q_{Y,K} = \mu_K^\pi(\bar{X}), \quad Q_{Y,j} = \mathbb{E}[Q_{Y,j+1}|\bar{X}_j, \bar{A}_j = \bar{\pi}_j(\bar{X}_j), \bar{C}_j = \mathbf{0}_j, \bar{S}_j = \mathbf{1}_j]
\end{aligned}$$

for $j = 1, \ldots, K - 1$. We adopt the point-mass contamination strategy as introduced in Hines et al. (2022) and Kennedy (2023). Let $\mathcal{P}$ represent the distribution of the observation $O$.

Consider the submodel $\mathcal{P}_t$, a perturbation of $\mathcal{P}$ in the direction of $\widetilde{\mathcal{P}}$ at the specific observation $\tilde{o} = \{\tilde{x}_1, \tilde{a}_1, \tilde{c}_1, (1 - \tilde{c}_1)\tilde{s}_1, (1 - \tilde{c}_1)\tilde{s}_1\tilde{x}_2, (1 - \tilde{c}_1)\tilde{s}_1\tilde{a}_2, (1 - \tilde{c}_1)\tilde{c}_2\tilde{s}_1, (1 - \tilde{c}_1)(1 - \tilde{c}_2)\tilde{s}_1\tilde{s}_2, (1 - \tilde{c}_1)(1 - \tilde{c}_2)\tilde{s}_1\tilde{s}_2\tilde{y}\}$. The Gâteaux derivative of the denominator $D(\mathcal{P}_t)$ evaluated at $t = 0$ can be derived as follows.

$$\frac{\partial}{\partial t} D(\mathcal{P}_t)\Big|_{t=0} \tag{6}$$

$$= \int p_1^0(x_1) \mathbb{E}(p_2^{00}(\bar{X}) \mid x_1, A_1 = C_1 = 0, S_1 = 1) \frac{\partial}{\partial t} f_t(x_1)\Big|_{t=0} dx_1 \tag{7}$$

$$+ \int \frac{\partial}{\partial t} \left( \int s_1 \frac{f_t(s_1, x_1, A_1 = C_1 = 0)}{f_t(x_1, A_1 = C_1 = 0)} ds_1 \right)\Big|_{t=0} \mathbb{E}(p_2^{00}(\bar{X}) | x_1, A_1 = C_1 = 0, S_1 = 1) f(x_1) dx_1 \tag{8}$$

$$+ \int p_1^0(x_1) \frac{\partial}{\partial t} \left( \int s_2 \frac{f_t(s_2, \bar{x}, \bar{A} = \bar{C} = \mathbf{0}_2, S_1 = 1)}{f_t(\bar{x}, \bar{A} = \bar{C} = \mathbf{0}_2, S_1 = 1)} \right. \tag{9}$$

$$\left. \times \frac{f_t(\bar{x}, A_1 = C_1 = 0, S_1 = 1)}{f_t(x_1, A_1 = C_1 = 0, S_1 = 1)} ds_2 dx_2 \right)\Big|_{t=0} f(x_1) dx_1 \tag{10}$$

$$= \int p_1^0(x_1) \mathbb{E}(p_2^{00}(\bar{X}) \mid x_1, A_1 = C_1 = 0, S_1 = 1) \{\mathbf{1}_{\tilde{x}_1}(x_1) - f(x_1)\} dx_1 \tag{11}$$

$$+ \int \frac{\mathbf{1}_{\tilde{x}_1}(x_1)(1 - \tilde{a}_1)(1 - \tilde{c}_1)}{f(x_1)\varphi^0(x_1)} \{\tilde{s}_1 - p_1^0(x_1)\} \mathbb{E}(p_2^{00}(\bar{X}) | x_1, A_1 = C_1 = 0, S_1 = 1) f(x_1) dx_1 \tag{12}$$

$$+ \int p_1^0(x_1) \frac{\mathbf{1}_{\tilde{x}_1}(x_1)}{f(x_1)} \left[ \frac{(1 - \tilde{a}_1)(1 - \tilde{a}_2)(1 - \tilde{c}_1)(1 - \tilde{c}_2)\tilde{s}_1}{\varphi_2^{00}(\tilde{x}_1, \tilde{x}_2)\varphi_1^0(\tilde{x}_1)p_1^0(\tilde{x}_1)} \{\tilde{s}_2 - p_2^{00}(\tilde{x}_1, \tilde{x}_2)\} \right. \tag{13}$$

$$\left. + \frac{(1 - \tilde{a}_1)(1 - \tilde{c}_1)\tilde{s}_1}{\varphi_1^0(\tilde{x}_1)p_1^0(\tilde{x}_1)} \{p_2^{00}(\tilde{x}_1, \tilde{x}_2) - m_{p_2}^{00}(\tilde{x}_1)\} \right] f(x_1) dx_1. \tag{14}$$

Consequently, the efficient influence function for $D$, denoted by $\psi_D(O)$, is given by $\psi_D(O) := \phi_D(O) - D(O)$.

Similarly, we have

$$\frac{\partial}{\partial t} N(\mathcal{P}_t)\Big|_{t=0} = \int m_{\mu_2}^{\pi}(x_1) p_1^0(x_1) m_{p_2}^{00}(x_1) \frac{\partial}{\partial t} f_t(x_1)\Big|_{t=0} dx_1 \tag{15}$$

$$+ \int m_{\mu_2}^{\pi}(x_1) p_1^0(x_1) \frac{\partial}{\partial t} m_{p_2}^{00}(x_1; \mathcal{P}_t)\Big|_{t=0} f(x_1) dx_1 \tag{16}$$

$$+ \int m_{\mu_2}^{\pi}(x_1) \frac{\partial}{\partial t} p_1^0(x_1; \mathcal{P}_t)\Big|_{t=0} m_{p_2}^{00}(x_1) f(x_1) dx_1 \tag{17}$$

$$+ \int \frac{\partial}{\partial t} m_{\mu_2}^{\pi}(x_1; \mathcal{P}_t)\Big|_{t=0} p_1^0(x_1) m_{p_2}^{00}(x_1) f(x_1) dx_1. \tag{18}$$

The simplification of each term yields

$$(15) = m_{\mu_2}^{\pi}(\tilde{x}_1)p_1^0(\tilde{x}_1)m_{p_2}^{00}(\tilde{x}_1) - N(\tilde{o}),$$

$$(16) = \int m_{\mu_2}^{\pi}(x_1)p_1^0(x_1)\frac{\mathbf{1}_{\tilde{x}_1}(x_1)}{f(x_1)}$$
$$\times \left[ \frac{(1-\tilde{a}_1)(1-\tilde{a}_2)(1-\tilde{c}_1)(1-\tilde{c}_2)\tilde{s}_1}{\varphi_2^{00}(\tilde{x}_1,\tilde{x}_2)\varphi_1^0(\tilde{x}_1)p_1^0(\tilde{x}_1)} \left\{ \tilde{s}_2 - p_2^{00}(\tilde{x}_1,\tilde{x}_2) \right\} \right.$$
$$\left. + \frac{(1-\tilde{a}_1)(1-\tilde{c}_1)\tilde{s}_1}{\varphi_1^0(\tilde{x}_1)p_1^0(\tilde{x}_1)} \left\{ p_2^{00}(\tilde{x}_1,\tilde{x}_2) - m_{p_2}^{00}(\tilde{x}_1) \right\} \right] f(x_1)dx_1,$$

$$(17) = \int m_{\mu_2}^{\pi}(x_1)\frac{\mathbf{1}_{\tilde{x}_1}(x_1)(1-\tilde{a}_1)(1-\tilde{c}_1)}{f(x_1)\varphi_1^0(x_1)} \left\{ \tilde{s}_1 - p_1^0(x_1) \right\} m_{p_2}^{00}(x_1)f(x_1)dx_1,$$

$$(18) = \int \left[ \frac{\mathbf{1}\{(\tilde{a}_1,\tilde{a}_2) = \pi(\tilde{x}_1,\tilde{x}_2)\}(1-\tilde{c}_1)(1-\tilde{c}_2)\tilde{s}_1\tilde{s}_2}{\varphi_2^{\bar{\tilde{a}}}(\tilde{x}_1,\tilde{x}_2)p_2^{\bar{\tilde{a}}}(\tilde{x}_1,\tilde{x}_2)\varphi_1^{\tilde{a}_1}(\tilde{x}_1)p_1^{\tilde{a}_1}(\tilde{x}_1)} \left\{ \tilde{y} - \mu_2^{\pi}(\tilde{x}_1,\tilde{x}_2) \right\} \right.$$
$$\left. + \frac{\mathbf{1}\{\tilde{a}_1 = \pi_1(\tilde{x}_1)\}(1-\tilde{c}_1)\tilde{s}_1}{\varphi_1^{\tilde{a}_1}(\tilde{x}_1)p_1^{\tilde{a}_1}(\tilde{x}_1)} \left\{ \mu_2^{\pi}(\tilde{x}_1,\tilde{x}_2) - m_{\mu_2}^{\pi}(\tilde{x}_1) \right\} \right]$$
$$\times \frac{\mathbf{1}_{\tilde{x}_1}(x_1)}{f(\tilde{x}_1)} \times p_1^0(x_1)m_{p_2}^{00}(x_1)f(x_1)dx_1.$$

Organizing the terms above yields the efficient influence function $\psi_N$ of $N$

$$\psi_N(O) = \phi_N(O) - N(O).$$

Finally, applying the product rule and the chain rule for Gâuteaux derivatives, we obtain

$$\psi_{V(\pi)} = \frac{1}{D}\psi_N - \frac{N}{D^2}\psi_D$$
$$= \frac{1}{D}(\phi_N - N) - \frac{1}{D}V(\pi)(\phi_D - D)$$
$$= \frac{1}{D}\left\{ \phi_N - V(\pi)\phi_D \right\}.$$

### A.1.3 Proof of Theorem 3

Let $\mathbb{P}\{f\}$ and $\mathbb{P}_n\{f\}$ denote the expectation of a function $f$ with respect to a probability measure $\mathbb{P}$ and its empirical counterpart $\mathbb{P}_n$, respectively. We assume the following regularity conditions (Assumption 9), which state positivity and boundedness for the parametric models.

**Assumption 9.**

(i) $\widehat{\varphi}_k^{\bar{a}_k}, \widehat{p}_k^{\bar{a}_k} \geq \epsilon$ a.s. for all $\bar{a}_k$ for some $\epsilon > 0$.

(ii) $|\widehat{\mu}_2^{a_1 a_2}(\bar{x})|, |\widehat{m}_{\mu_2}^{a_1 a_2}(x_1)| < L$ for all $\bar{x}, \bar{a}$ for some $L < \infty$.

*Proof of Theorem 3.* Denote $\theta^* = \{\alpha_k^*, \eta_k^*, \gamma_k^*, \zeta^*, \xi^*, \nu^* : k = 1, 2\}$ as the probability limits of $\hat{\theta} = \{\hat{\alpha}_k, \hat{\eta}_k, \hat{\gamma}_k, \hat{\zeta}, \hat{\xi}, \hat{\nu} : k = 1, 2\}$. For simplicity, define $\widehat{e}_k(\cdot) = e_k(\cdot; \hat{\alpha}_k)$, $e_k^*(\cdot) = e_k(\cdot; \alpha_k^*)$, and similarly for other nuisance models. Additionally, let

$$m_{p_2^*}^{a_1 a_2}(x_1) = \mathbb{E}(p_2^{*a_1 a_2}(\bar{X})|x_1, a_1, C_1 = 0, S_1 = 1),$$
$$m_{\mu_2^*}^{a_1 a_2}(x_1) = \mathbb{E}(\mu_2^{*a_1 a_2}(\bar{X})|x_1, a_1, C_1 = 0, S_1 = 1).$$

To demonstrate the multiple robustness of $\widehat{V}(\pi)$, it is sufficient to show that (i) $\mathbb{E}\phi_D^* = D$ and (ii) $\mathbb{E}\phi_N^* = N$.

(i)

$$\mathbb{E}[\phi_D^*] = \mathbb{E}\Bigg[\frac{(1-A_1)(1-C_1)}{\varphi_1^{*0}(X_1)}\left\{S_1 - p_1^{*0}(X_1)\right\} m_{p_2}^{*00}(X_1)$$

$$+ \Bigg[\frac{(1-A_1)(1-A_2)(1-C_1)(1-C_2)S_1}{\varphi_1^{*0}(X_1)p_1^{*0}(X_1)\varphi_2^{*00}(\bar{X})}\left(S_2 - p_2^{*00}(\bar{X})\right)$$

$$+ \frac{(1-A_1)(1-C_1)S_1}{\varphi_1^{*0}(X_1)p_1^{*0}(X_1)}\left\{p_2^{*00}(\bar{X}) - m_{p_2}^{*00}(X_1)\right\}\Bigg] \times p_1^{*0}(X_1)$$

$$+ p_1^{*0}(X_1)m_{p_2}^{*00}(X_1)\Bigg]$$

$$= \mathbb{E}\left[\frac{\varphi_1^0(X_1)}{\varphi_1^{*0}(X_1)}\left\{p_1^0(X_1) - p_1^{*0}(X_1)\right\} m_{p_2}^{*00}(X_1)\right] + \mathbb{E}\left[p_1^{*0}(X_1)m_{p_2}^{*00}(X_1)\right]$$

$$+ \mathbb{E}\Bigg(\Bigg[\mathbb{E}\left\{\frac{\varphi_2^{00}(\bar{X})}{\varphi_2^{*00}(\bar{X})}\left(p_2^{00}(\bar{X}) - p_2^{*00}(\bar{X})\right)\Big|X_1, A_1 = C_1 = 0, S_1 = 1\right\}$$

$$+ \left\{m_{p_2^*}^{00}(X_1) - m_{p_2}^{*00}(X_1)\right\}\Bigg] \times p_1^0(X_1)\frac{\varphi_1^0(X_1)}{\varphi_1^{*0}(X_1)}\Bigg)$$

If any of the following scenarios is met,

(i-1) $\widehat{\varphi}_1, \widehat{\varphi}_2$ are correctly specified.
(i-2) $\widehat{\varphi}_1, \widehat{p}_2$ are correctly specified.
(i-3) $\widehat{p}_1, \widehat{p}_2, \widehat{m}_{p_2}$ are correctly specified.

then $\mathbb{P}_n\{\hat{\phi}_D\}$ converges in probability to $D$.

(ii) Similarly, by the law of iterated expectation,

$$\mathbb{E}\phi_N^* \tag{19}$$

$$= \mathbb{E}\Bigg[\Bigg(\frac{\mathbf{1}\{\pi(\bar{X}) = \bar{A}\}(1-C_1)(1-C_2)S_1S_2}{\varphi_1^{*a_1}(X_1)p_1^{*a_1}(X_1)\varphi_2^{*\bar{a}}(\bar{X})p_2^{*\bar{a}}(\bar{X})}\left\{Y - \mu_2^{*\pi}(\bar{X})\right\} \tag{20}$$

$$+ \frac{\mathbf{1}\{\pi_1(X_1) = A_1\}(1-C_1)S_1}{\varphi_1^{*a_1}(X_1)p_1^{*a_1}(X_1)}\left\{\mu_2^{*\pi}(\bar{X}) - m_{\mu_2}^{*\pi}(X_1)\right\}\Bigg) \tag{21}$$

$$\times p_1^{*0}(X_1)m_{p_2}^{*00}(X_1) \tag{22}$$

$$+ m_{\mu_2}^{*\pi}(X_1)\left[\frac{(1-A_1)(1-C_1)}{\varphi_1^{*0}(X_1)}\left(S_1 - p_1^{*0}(X_1)\right)\right] \cdot m_{p_2}^{*00}(X_1) \tag{23}$$

$$+ \Bigg[\frac{(1-A_1)(1-A_2)(1-C_1)(1-C_2)S_1}{\varphi_1^{*0}(X_1)p_1^{*0}(X_1)\varphi_2^{*00}(\bar{X})}\left(S_2 - p_2^{*00}(\bar{X})\right) \tag{24}$$

$$+ \frac{(1-A_1)(1-C_1)S_1}{\varphi_1^{*0}(X_1)p_1^{*0}(X_1)}\left\{p_2^{*00}(\bar{X}) - m_{p_2}^{*00}(X_1)\right\}\Bigg] m_{\mu_2}^{*\pi}(X_1)p_1^{*0}(X_1) \tag{25}$$

$$+ m_{\mu_2}^{*\pi}(X_1)p_1^{*0}(X_1)m_{p_2}^{*00}(X_1)\Bigg] \tag{26}$$

$$= \mathbb{E}\Bigg[\frac{\varphi_1^{\pi_1}(X_1)p_1^{\pi_1}(X_1)}{\varphi_1^{*\pi_1}(X_1)p_1^{*\pi_1}(X_1)}\Bigg\{\left(m_{\mu_2^*}^{\pi}(X_1) - m_{\mu_2}^{*\pi}(X_1)\right) \tag{27}$$

$$+ \mathbb{E}\left(\frac{\varphi_2^{\pi}(\bar{X})p_2^{\pi}(\bar{X})}{\varphi_2^{*\pi}(\bar{X})p_2^{*\pi}(\bar{X})}\left(\mu_2^{\pi}(\bar{X}) - \mu_2^{*\pi}(\bar{X})\right)\Big|X_1, A_1 = \pi_1(X_1), C_1 = 0, S_1 = 1\right)\Bigg\} \tag{28}$$

$$\times p_1^{*0}(X_1)m_{p_2}^{*00}(X_1) + m_{\mu_2}^{*\pi}(X_1)\phi_D^* \tag{29}$$

$$+ m_{\mu_2}^{*\pi}(X_1)p_1^{*0}(X_1)m_{p_2}^{*00}(X_1)\Bigg]. \tag{30}$$

To enumerate the scenarios for a consistent value estimator, we only consider those with a consistent denominator.

(ii-1) $\widehat{p}_1, \widehat{p}_2, \widehat{m}_{p_2}$ are correctly specified

(ii-2) $\widehat{p}_1, \widehat{m}_{p_2}, \widehat{\mu}_2$ are correctly specified

(ii-3) $\widehat{\mu}_2, \widehat{m}_{\mu_2}$ are correctly specified

If (i-1) is met, we also need (ii-1) or (ii-3) for $\mathbb{E}\phi_N^* = N$. If (i-2) is met, then (ii-2) or (ii-3) is required, and if (i-3) is met, then (ii-3) is required.

Consequently, $\widehat{V}(\pi)$ achieves consistency if the scenario is within:

$\{\ \widehat{\varphi}_1, \widehat{p}_1, \widehat{\varphi}_2, \widehat{p}_2, \widehat{m}_{p_2}$ are correctly specified $\} \cup \{\ \widehat{\varphi}_1, \widehat{\varphi}_2, \widehat{\mu}_2, \widehat{m}_{\mu_2}$ are correctly specified $\}$

$\cup \{\ \widehat{\varphi}_1, \widehat{p}_1, \widehat{p}_2, \widehat{m}_{p_2}, \widehat{\mu}_2$ are correctly specified $\} \cup \{\ \widehat{\varphi}_1, \widehat{p}_2, \widehat{\mu}_2, \widehat{m}_{\mu_2}$ are correctly specified $\}$

$\cup \{\ \widehat{p}_1, \widehat{p}_2, \widehat{m}_{p_2}, \widehat{\mu}_2, \widehat{m}_{\mu_2}$ are correctly specified $\}$.

Now, we will show that if all nuisance models are correctly specified, $\widehat{V}(\pi)$ has $\psi_{V(\pi)}$ as its efficient influence function. We use a dot to denote the partial derivative with respect to the model parameter $\theta$, for example, $\dot{N}^*(O) = \dot{N}(O;\theta^*) = \left.\frac{\partial N(O;\theta)}{\partial\theta}\right|_{\theta=\theta^*}$.

By Taylor expansion at $\theta^*$, we have, for general functionals $N(O)$ and $D(O)$,

$$\mathbb{P}_n\{N(O;\hat{\theta})\} = \mathbb{P}_n\{N(O;\theta^*)\} + \mathbb{P}\{\dot{N}(O;\theta^*)\}(\hat{\theta} - \theta^*) + o_{\mathbb{P}}(n^{-1/2}), \tag{31}$$

$$\mathbb{P}_n\{D(O;\hat{\theta})\} = \mathbb{P}_n\{D(O;\theta^*)\} + \mathbb{P}\{\dot{D}(O;\theta^*)\}(\hat{\theta} - \theta^*) + o_{\mathbb{P}}(n^{-1/2}), \tag{32}$$

$$\frac{\mathbb{P}_n\{N(O;\hat{\theta})\}}{\mathbb{P}_n\{D(O;\hat{\theta})\}} = \frac{\mathbb{P}_n\{N(O;\hat{\theta})\}}{\mathbb{P}\{D(O;\theta^*)\}} - \frac{\mathbb{P}\{N(O;\theta^*)\}}{[\mathbb{P}\{D(O;\theta^*)\}]^2}\left[\mathbb{P}_n\{D(O;\hat{\theta})\} - \mathbb{P}\{D(O;\theta^*)\}\right] + o_{\mathbb{P}}(n^{-1/2}). \tag{33}$$

Plugging in $N(O;\theta) = \phi_N(\theta)$ and $D(O;\theta) = \phi_D(\theta)$ results in

$$\widehat{V}(\pi) - V(\pi)$$

$$= \frac{\mathbb{P}_n\{\hat{\phi}_N\}}{\mathbb{P}\{\phi_D^*\}} - \frac{\mathbb{P}\{\phi_N^*\}}{[\mathbb{P}\{\phi_D^*\}]^2}\left[\mathbb{P}_n\{\hat{\phi}_D\} - \mathbb{P}\{\phi_D^*\}\right] - V(\pi) + o_{\mathbb{P}}(n^{-1/2}) \qquad \text{(by (33))}$$

$$= \mathbb{P}_n\left\{\frac{1}{\mathbb{P}\{\phi_D^*\}}\left(\hat{\phi}_N - \frac{\mathbb{P}\{\phi_N^*\}}{\mathbb{P}\{\phi_D^*\}}\hat{\phi}_D\right)\right\} + o_{\mathbb{P}}(n^{-1/2})$$

$$= \mathbb{P}_n\left\{\frac{1}{p_2^{00}}\left(\hat{\phi}_N - V(\pi)\hat{\phi}_D\right)\right\} + o_{\mathbb{P}}(n^{-1/2})$$

$$= \mathbb{P}_n\left\{\frac{1}{p_2^{00}}\left((\phi_N^* + \mathbb{P}\{\dot{\phi}_N^*\})(\hat{\theta} - \theta^*)) - V(\pi)\left(\phi_D^* + \mathbb{P}\{\dot{\phi}_D^*\}(\hat{\theta} - \theta^*)\right)\right)\right\} + o_{\mathbb{P}}(n^{-1/2})$$

$$\text{(by (31), (32))}$$

$$= \mathbb{P}_n\left\{\frac{1}{p_2^{00}}\left(\phi_N^* - V(\pi)\phi_D^*\right)\right\} + o_{\mathbb{P}}(n^{-1/2})$$

$$= \mathbb{P}_n\{\psi_{V(\pi)}\} + o_{\mathbb{P}}(n^{-1/2}). \tag{*}$$

The penultimate equality follows because $\mathbb{P}\{\dot{\phi}_N^*\} = \mathbb{P}\{\dot{\phi}_D^*\} = 0$. $\qquad\square$

### A.1.4  Proof of Theorem 4

Let $\|\cdot\|_P$ denote the $L^2(P)$-norm with respect to a probability measure $P$. We omit the subscript when the corresponding measure is clear from the context. Regularity condition 10 states that the nonparametric models satisfy positivity and boundedness and are consistent with the true data-generating models, similar to Assumption 9. It additionally assumes that the rate of convergence for each model does not exceed a certain level.

**Assumption 10.**

1. $\widehat{\varphi}_k^{\bar{a}_k}, \widehat{p}_k^{\bar{a}_k} \geq \epsilon$ a.s. $k = 1, 2,$ for some $\epsilon > 0$.

2. $|\widehat{\mu}_2^{a_1 a_2}(\bar{x})|, |\widehat{m}_{\mu_2}^{a_1 a_2}(x_1)| < L$ for all $\bar{x}, \bar{a}$ for some $L < \infty$.

3. $\hat{\theta} \xrightarrow{\mathbb{P}} \theta$ as $n \to \infty$.

4. $\|\hat{g} - g\| \cdot \|\hat{h} - h\| = o_P(n^{-1/2})$ for all $g \neq h$ where $g, h \in \{\varphi_k^{\bar{a}_k}, p_k^{\bar{a}_k}, \mu_2^{a_1 a_2}, m_{p_2}^{00}, m_{\mu_2}^{a_1 a_2} :$ $\bar{a} \in \{0, 1\}^2, k = 1, 2\}$.

*Proof of Theorem 4.* It suffices to show that the EIF of $\widehat{V}(\pi)$ is $\psi_{V(\pi)}$. This means showing that

$$\widehat{V}(\pi) - V(\pi) = \mathbb{P}_n\{\psi_{V(\pi)}\} + o_{\mathbb{P}}(n^{-1/2}).$$

1. We observe that, by algebra,

$$\mathbb{P}_n\{\hat{\phi}_D\} - \mathbb{P}\{\phi_D^*\}$$
$$= (\mathbb{P}_n - \mathbb{P})\{\phi_D^*\} + (\mathbb{P}_n - \mathbb{P})\{\hat{\phi}_D - \phi_D^*\} + \mathbb{P}\{\hat{\phi}_D - \phi_D^*\}.$$

The first term is the centered empirical mean, which will converge to a normal distribution as $n \to \infty$. The second term can be simplified to

$$(\mathbb{P}_n - \mathbb{P})\{\hat{\phi}_D - \phi_D^*\} = \mathcal{O}_{\mathbb{P}}\left(\frac{\|\hat{\phi}_D - \phi_D^*\|}{\sqrt{n}}\right) = o_{\mathbb{P}}(n^{-1/2})$$

by Lemma 1 from Kennedy (2023), where the last equality follows from the bounded convergence theorem because $\hat{\phi}_D \xrightarrow{\mathbb{P}} \phi_D^*$ and $|\phi_D(\theta)|$ is bounded almost surely.

For simplicity, let

$$m_{\widehat{p}_2}^{a_1 a_2}(x_1) = \mathbb{E}[\widehat{p}_2^{a_1 a_2}(\bar{X}) \mid x_1, a_1, 0, 1],$$
$$m_{\widehat{\mu}, \pi}^{a_1 a_2}(x_1) = \mathbb{E}[\widehat{\mu}_2^{\pi}(\bar{X}) \mid x_1, \pi_1, 0, 1]$$

denote the conditional means of the pseudo-outcomes. To bound the convergence rate of the third term $T_D := \mathbb{P}\{\hat{\phi}_D - \phi_D^*\}$, we observe that by the law of iterated expectation,

$$T_D = \mathbb{E}\left[\frac{\varphi_1^0(X_1)}{\widehat{\varphi}_1^0(X_1)}\left\{p_1^0(X_1) - \widehat{p}_1^0(X_1)\right\}\widehat{m}_{p_2}^{00}(X_1)\right.$$
$$+ \widehat{p}_1^0(X_1)\frac{\varphi_1^0(X_1)p_1^0(X_1)}{\widehat{\varphi}_1^0(X_1)\widehat{p}_1^0(X_1)}\left(\mathbb{E}\left[\frac{\varphi_2^{00}(\bar{X})}{\widehat{\varphi}_2^{00}(\bar{X})}\left\{p_2^{00}(\bar{X}) - \widehat{p}_2^{00}(\bar{X})\right\}|X_1, 0, 0, 1\right]\right.$$
$$\left.+ \left\{m_{\widehat{p}_2}^{00}(X_1) - \widehat{m}_{p_2}^{00}(X_1)\right\}\right)$$
$$\left.+ \widehat{p}_1^0(X_1)\widehat{m}_{p_2}^{00}(X_1) - p_1^0(X_1)m_{p_2}^{00}(X_1)\right]$$
$$= \mathbb{E}\left[\frac{\widehat{\varphi}_1^0(X_1) - \varphi_1^0(X_1)}{\widehat{\varphi}_1^0(X_1)}\widehat{p}_1^0(X_1)\left\{\widehat{m}_{p_2}^{00}(X_1) - m_{\widehat{p}_2}^{00}(X_1)\right\}\right.$$
$$+ \frac{\widehat{\varphi}_1^0(X_1) - \varphi_1^0(X_1)}{\widehat{\varphi}_1^0(X_1)}\left\{\widehat{p}_1^0(X_1) - p_1^0(X_1)\right\}m_{\widehat{p}_2}^{00}(X_1)$$
$$+ \mathbb{E}\left(\left\{p_1^0(X_1)\left(\frac{\varphi_1^0(X_1)}{\widehat{\varphi}_1^0(X_1)}\left(\frac{\varphi_2^{00}(\bar{X})}{\widehat{\varphi}_2^{00}(\bar{X})} - 1\right) + \left(\frac{\varphi_1^0(X_1)}{\widehat{\varphi}_1^0(X_1)} - 1\right)\right)\right.\right.$$
$$\left.\left.+ (p_1^0(X_1) - \widehat{p}_1^0(X_1))\right\} \times \{p_2^{00}(\bar{X}) - \widehat{p}_2^{00}(\bar{X})\}|X_1, 0, 0, 1\right)$$
$$\left.+ \left\{p_1^0(X_1) - \widehat{p}_1^0(X_1)\right\}\left\{m_{\widehat{p}_2}^{00}(X_1) - m_{p_2}^{00}(X_1)\right\}\right].$$

The second equality is obtained by rearranging the terms, which involves adding and subtracting the same quantities. By the Cauchy-Schwarz inequality and conditional Jensen's inequality, we have, for some constant $K_D$,

$$|T_D| \le K_D \times \left[ \|\widehat{\varphi}_1^0 - \varphi_1^0\| \cdot \left( \|\widehat{m}_{p_2}^{00} - m_{p_2}^{00}\| + \|\widehat{p}_1^0 - p_1^0\| \right) \right.$$
$$+ \left( \|\widehat{\varphi}_2^{00} - \varphi_2^{00}\| + \|\widehat{\varphi}_1^0 - \varphi_1^0\| + \|\widehat{p}_1^0 - p_1^0\| \right) \cdot \|\widehat{p}_2^{00} - p_2^{00}\|$$
$$\left. + \|\widehat{p}_1^0 - p_1^0\| \cdot \|m_{\widehat{p}_2}^{00} - m_{p_2}^{00}\| \right] = o_{\mathbb{P}}(n^{-1/2}).$$

2. By the same argument, we only need to determine the rate of convergence for the term $T_N := \mathbb{P}\{\hat{\phi}_N - \phi_N^*\}$. Again, by rearranging the terms and using the fact that $T_D = o_{\mathbb{P}}(n^{-1/2})$, we have

$$T_N$$
$$= \mathbb{E}\left[ \frac{\varphi_1^{\pi_1}(X_1) p_1^{\pi_1}(X_1)}{\widehat{\varphi}_1^{\pi_1}(X_1) \widehat{p}_1^{\pi_1}(X_1)} \left[ \left\{ m_{\hat{\mu}_2}^{\pi}(X_1) - \widehat{m}_{\mu_2}^{\pi}(X_1) \right\} \right. \right.$$
$$+ \mathbb{E}\left\{ \frac{\varphi_2^{\pi}(\bar{X}) p_2^{\pi}(\bar{X})}{\widehat{\varphi}_2^{\pi}(\bar{X}) \widehat{p}_2^{\pi}(\bar{X})} \{\mu_2^{\pi}(\bar{X}) - \hat{\mu}_2^{\pi}(\bar{X})\} \Big| X_1, A_1 = \pi_1(X_1), C_1 = 0, S_1 = 1 \right\} \right]$$
$$\times \widehat{p}_1^0(X_1) \widehat{m}_{p_2}^{00}(X_1) + \widehat{m}_{\mu_2}^{\pi}(X_1) \widehat{\phi}_D - m_{\mu_2}^{\pi}(X_1) p_1^0(X_1) m_{p_2}^{00}(X_1) \right]$$
$$= \mathbb{E}\left[ \frac{\varphi_1^{\pi_1}(X_1) p_1^{\pi_1}(X_1)}{\widehat{\varphi}_1^{\pi_1}(X_1) \widehat{p}_1^{\pi_1}(X_1)} \left[ \left\{ m_{\hat{\mu}_2}^{\pi}(X_1) - \widehat{m}_{\mu_2}^{\pi}(X_1) \right\} \right. \right.$$
$$+ \mathbb{E}\left\{ \frac{\varphi_2^{\pi}(\bar{X}) p_2^{\pi}(\bar{X})}{\widehat{\varphi}_2^{\pi}(\bar{X}) \widehat{p}_2^{\pi}(\bar{X})} \{\mu_2^{\pi}(\bar{X}) - \hat{\mu}_2^{\pi}(\bar{X})\} \Big| X_1, A_1 = \pi_1(X_1), C_1 = 0, S_1 = 1 \right\} \right]$$
$$\times \widehat{p}_1^0(X_1) \widehat{m}_{p_2}^{00}(X_1) + \{\widehat{m}_{\mu_2}^{\pi}(X_1) - m_{\mu_2}^{\pi}(X_1)\} p_1^0(X_1) m_{p_2}^{00}(X_1) \right] + o_{\mathbb{P}}(n^{-1/2})$$
$$= \mathbb{E}\left[ \left( \frac{\varphi_1^{\pi_1}(X_1)}{\widehat{\varphi}_1^{\pi_1}(X_1)} - 1 \right) \left\{ m_{\hat{\mu}_2}^{\pi}(X_1) - \widehat{m}_{\mu_2}^{\pi}(X_1) \right\} p_1^0(X_1) \widehat{m}_{p_2}^{00}(X_1) \right.$$
$$+ \left( \frac{\varphi_1^{\pi_1}(X_1)}{\widehat{\varphi}_1^{\pi_1}(X_1)} - 1 \right) \mathbb{E}\left\{ \frac{\varphi_2^{\pi}(\bar{X}) p_2^{\pi}(\bar{X})}{\widehat{\varphi}_2^{\pi}(\bar{X}) \widehat{p}_2^{\pi}(\bar{X})} \{\mu_2^{\pi}(\bar{X}) - \hat{\mu}_2^{\pi}(\bar{X})\} \Big| X_1, \pi_1, 0, 1 \right\} p_1^0(X_1) \widehat{m}_{p_2}^{00}(X_1)$$
$$+ \{\widehat{m}_{\mu_2}^{\pi}(X_1) - m_{\hat{\mu}_2}^{\pi}(X_1)\} \{p_1^0(X_1) m_{p_2}^{00}(X_1) - \widehat{p}_1^0(X_1) \widehat{m}_{p_2}^{00}(X_1)\}$$
$$\left. + \{m_{\hat{\mu}_2}^{\pi}(X_1) - m_{\mu_2}^{\pi}(X_1)\} \{p_1^0(X_1) m_{p_2}^{00}(X_1) - \widehat{p}_1^0(X_1) \widehat{m}_{p_2}^{00}(X_1)\} \right] + o_{\mathbb{P}}(n^{-1/2}).$$

By the Cauchy-Schwarz and conditional Jensen's inequalities, we obtain:

$$|T_N| \le K_N \times \left[ \|\varphi_1^{\pi_1} - \widehat{\varphi}_1^{\pi_1}\| \left( \|\mu_2^{\pi} - \widehat{\mu}_2^{\pi}\| + \|m_{\mu_2}^{\pi} - \widehat{m}_{\mu_2}^{\pi}\| \right) \right.$$
$$\left. + \left( \|\mu_2^{\pi} - \widehat{\mu}_2^{\pi}\| + \|m_{\mu_2}^{\pi} - \widehat{m}_{\mu_2}^{\pi}\| \right) \left( \|p_1^0 - \widehat{p}_1^0\| + \|m_{p_2}^{00} - \widehat{m}_{p_2}^{00}\| \right) \right]$$
$$+ o_{\mathbb{P}}(n^{-1/2}) = o_{\mathbb{P}}(n^{-1/2})$$

for some constant $K_N$.

By 1 and 2, we have (31), (32), and (33). Hence, the desired result follows. $\qquad\square$

### A.1.5 Proof of Theorem 5

Let $V'(\beta^*)$ and $V''(\beta^*)$ denote the first and second-order derivatives of $V(\beta)$ evaluated at $\beta^*$, respectively. In addition to Assumption 10, We further assume the following regularity conditions, which guarantee a well-defined optimization problem.

**Assumption 11.**

(i) $V(\beta)$ is twice continuously differentiable in the neighborhood of $\beta^*$.

(ii) There exists $\delta_0 > 0$ such that $\mathbb{P}(|\widetilde{H}_k^\mathsf{T}\beta_k| \leq \delta) = O(\delta)$, $k = 1, 2$ uniformly in $0 \leq \delta \leq \delta_0$.

(iii) $V(\beta; \theta)$ is Fréchet differentiable at $\theta^*$, and the Fréchet derivative $|\dot{V}(\beta; \theta^*)| \leq M$ for almost all $o$ and $\beta$ for some $0 < M < \infty$.

The first condition is a standard regularity condition, ensuring the smoothness of the objective surface. This condition is essential for well-behaved optimization procedures and guarantees uniform convergence of the estimators. The second condition, a margin condition (Luedtke and Van Der Laan, 2016), ensures that the probability of the undecidable boundary case, where $|\tilde{H}_k^\mathsf{T}\beta_k^*| = 0$, is zero. The third condition imposes a local smoothness on the always-survivor value function around the true data generating mechanism $\theta^*$. This condition prevents abrupt changes in the value function in the neighborhood of $\theta^*$.

The following lemma dictates the rate of convergence of $\hat{\beta}$ to $\beta^*$, thus will be useful when proving asymptotic normality of $\widehat{V}_{\text{AS}}(\hat{\beta})$.

**Lemma 2.** *Under assumptions 1, 3, 10, and 11, we have $n^{1/3}\|\hat{\beta} - \beta^*\| = O_\mathbb{P}(1)$.*

*Proof of Lemma 2.* The proof proceeds in two steps. First, we will show that $\beta^*$ is the probability limit of $\hat{\beta}$ by applying the argmax theorem.

(i) $V(\beta)$ is twice continuously differentiable at a neighborhood of $\beta^*$ by the assumption 9.

(ii) By the proof of Theorem 4, $\widehat{V}(\beta)$ is consistent to $V(\beta)$ for any $\beta$.

(iii) $\widehat{V}(\hat{\beta}) \geq \sup_{\beta:\|\beta\|=1} \widehat{V}(\beta)$ by definition of $\hat{\beta}$.

As the requisite conditions are met, it follows that $\hat{\beta}$ converges in probability to $\beta^*$ as $n$ approaches infinity.

The second stage of the proof employs Theorem 14.4 in conjunction with Lemma 9.6, Lemma 9.9, and Theorem 11.1 as presented in Kosorok (2008). The verification of the subsequent three conditions is performed:

(i) Consider a constant $\epsilon > 0$, and select $\beta$ such that the norm $\|\beta - \beta^*\| < \epsilon$. Given that $V'(\beta^*) = 0$, the second-order Taylor expansion of $V(\beta)$ in the vicinity of $\beta^*$ is given by:

$$V(\beta) - V(\beta^*) = \frac{1}{2}V''(\beta^*)\|\beta - \beta^*\|^2 + o(\|\beta - \beta^*\|^2).$$

Since $\beta^*$ is a point of maximum and $V(\beta)$ is twice differentiable, $V(\beta)$ is strictly concave around $\beta^*$ without loss of generality. Thus, there exists a constant $c_1 > 0$ such that $-\frac{1}{2}V''(\beta^*) \geq c_1$. The second-order Taylor expansion of $V(\beta)$ at $\beta^*$ then yields $V(\beta) - V(\beta^*) < (-c_1 + \delta)\|\beta - \beta^*\|^2$ for some $\delta > 0$ and $\beta$ in a sufficiently small neighborhood of $\beta^*$.

(ii) We want to show that for some constant $c_2 > 0$ and a function $\phi_n(\epsilon)$ such that the ratio $\phi_n(\epsilon)/\epsilon^\alpha$ does not depend on $n$ for some $\alpha < 2$, we have:

$$\mathbb{E}^*\left[n^{1/2}\sup_{\|\beta - \beta^*\|<\epsilon}\left|\widehat{V}(\beta) - V(\beta) - \left\{\widehat{V}(\beta^*) - V(\beta^*)\right\}\right|\right] \leq c_2\phi_n(\epsilon), \qquad (34)$$

where $\mathbb{E}^*(U) = \inf\{\mathbb{E}(U) : X \geq U \text{ is a random variable}, -\infty \leq \mathbb{E}(U) \leq \infty \text{ exists}\}$ denotes the outer expectation. We claim that $\phi_n(\epsilon) = \epsilon^{1/2} + \epsilon$ and $\alpha = 3/2$ satisfies the condition.

By Theorem 4, we have, for any $\beta$,

$$\widehat{V}(\beta) - V(\beta) = \mathbb{P}_n\left\{\frac{\phi_{N;\beta} - V(\beta)\phi_D}{D}\right\} + o_{\mathbb{P}}(n^{-1/2}),$$

and that the trailing $o_{\mathbb{P}}(n^{-1/2})$ term is bounded by a quantity proportional to $\|\hat{\theta} - \theta^*\|^2 + (\mathbb{P}_n - \mathbb{P})\{\dot{V}(\theta^*)\}\|\hat{\theta} - \theta^*\|$. It follows that

LHS of $(34)$

$$= \mathbb{E}^*\left[n^{1/2}\sup_{\|\beta-\beta^*\|<\epsilon}\left|\mathbb{P}_n\left\{\frac{\phi_{N;\beta} - V(\beta)\phi_D}{D}\right\} - \mathbb{P}_n\left\{\frac{\phi_{N;\beta^*} - V(\beta^*)\phi_D}{D}\right\} + o_{\mathbb{P}}(n^{-1/2})\right|\right]$$

$$= \mathbb{E}^*\left[n^{1/2}\sup_{\|\beta-\beta^*\|<\epsilon}\left|\mathbb{P}_n\left\{\frac{\phi_{N;\beta} - V(\beta)\phi_D + V(\beta)D}{D} - V(\beta)\right\}\right.\right.$$

$$\left.\left. - \mathbb{P}_n\left\{\frac{\phi_{N;\beta^*} - V(\beta^*)\phi_D + V(\beta^*)D}{D} - V(\beta^*)\right\} + o_{\mathbb{P}}(n^{-1/2})\right|\right]$$

$$= \underbrace{\mathbb{E}^*\left[n^{1/2}\sup_{\|\beta-\beta^*\|<\epsilon}\left|\mathbb{P}_n\left\{\frac{\mathbb{P}_n\{\phi_{N;\beta} - \phi_{N;\beta^*}\}}{D} + \{V(\beta) - V(\beta^*)\}\right\}\right|\right]}_{=:\tau_1}$$

$$- \underbrace{\mathbb{E}^*\left[n^{1/2}\sup_{\|\beta-\beta^*\|<\epsilon}\left|\{V(\beta) - V(\beta^*)\}\left\{1 - \frac{\mathbb{P}_n\{\phi_D\}}{D}\right\}\right|\right]}_{=:\tau_2} + o_{\mathbb{P}}(1).$$

Note that

$$\phi_{N;\beta} - \phi_{N;\beta*} \in \mathcal{F}_\beta(\bar{x}, \bar{a}, \bar{c}, \bar{s}, \bar{y})$$

$$= \left\{d_{\beta,\beta^*}(\phi_N^{a_1 a_2}) : \|\beta - \beta^*\| < \epsilon\right\}$$

where

$$\phi_N^{a_1 a_2}(O) := (\psi_{\mu_2}^{a_1 a_2}(O) + \psi_{m_\mu}^{a_1 a_2}(O))p_1^0(X_1)m_{p_2}^{00}(X_1)$$
$$+ m_{\mu_2}^{a_1 a_2}(X_1)\psi_{p_1}^0(O)m_{p_2}^{00}(X_1)$$
$$+ m_{\mu_2}^{a_1 a_2}(X_1)p_1^0(X_1)(\psi_{p_2}^{00}(O) + \psi_{m_{p_2}}^{00}(O)),$$

$$d_{\beta,\beta^*}(Z^{a_1 a_2}) := (Z^{11} - Z^{10} - Z^{01} + Z^{00})\left(\mathbf{1}\{\tilde{X}_1^\intercal\beta_1, \tilde{X}_2^\intercal\beta_2 > 0\} - \mathbf{1}\{\tilde{X}_1^\intercal\beta_1^*, \tilde{X}_2^\intercal\beta_2^* > 0\}\right)$$

$$+ (Z^{10} - Z^{00})\left(\mathbf{1}\{\tilde{X}_1^\intercal\beta_1 > 0\} - \mathbf{1}\{\tilde{X}_1^\intercal\beta_1^* > 0\}\right)$$

$$+ (Z^{01} - Z^{00})\left(\mathbf{1}\{\tilde{X}_2^\intercal\beta_2 > 0\} - \mathbf{1}\{\tilde{X}_2^\intercal\beta_2^* > 0\}\right)$$

for a potential outcome $Z^{a_1 a_2}$, and

$$\psi_{p_1}^0(\bar{X}) := \frac{(1 - A_1)(1 - C_1)}{\varphi_1^0(X_1)} \left\{ S_1 - p_1^0(X_1) \right\},$$

$$\psi_{p_2}^{00}(\bar{X}) := \prod_{k=1}^{2} \frac{(1 - A_k)(1 - C_k)S_{k-1}}{\varphi_k^{\bar{0}_k}(\bar{X}_k)p_{k-1}^{\bar{0}_k}(\bar{X}_k)} \left\{ S_2 - p_2^{00}(\bar{X}) \right\},$$

$$\psi_{\mu_2}^{a_1 a_2}(\bar{X}) := \prod_{k=1}^{2} \frac{\mathbf{1}\{A_k = a_k\}(1 - C_k)S_{k-1}}{\varphi_k^{\bar{a}_k}(\bar{X}_k)p_{k-1}^{\bar{a}_k}(\bar{X}_k)} \frac{S_2}{p_2^{a_1 a_2}(\bar{X})} \left\{ Y - \mu_2^{a_1 a_2}(\bar{X}) \right\},$$

$$\psi_{m_\mu}^{a_1 a_2}(\bar{X}) := \frac{\mathbf{1}\{A_1 = a_1\}(1 - C_1)S_1}{\varphi_1^{a_1}(X_1)p_1^{a_1}(X_1)} \left\{ \mu_2^{a_1 a_2}(\bar{X}) - m_{\mu_2}^{a_1 a_2}(X_1) \right\},$$

$$\psi_{m_{p_2}}^{00}(\bar{X}) := \frac{(1 - A_1)(1 - C_1)S_1}{\varphi_1^0(X_1)p_1^0(X_1)} \left\{ p_2^{00}(\bar{X}) - m_{p_2}^{00}(X_1) \right\}.$$

With a slight abuse of notation, define

$$d_{11}(Z) := Z^{11} - Z^{10} - Z^{01} + Z^{00},$$
$$d_{10}(Z) = Z^{10} - Z^{00},$$
$$d_{01}(Z) = Z^{01} - Z^{00}.$$

By Assumption 6, $M = \sup_o |d_{11}(\phi_N(o))| + \sup_o |d_{10}(\phi_N(o))| + \sup_o |d_{01}(\phi_N(o))| < \infty$.

For $\|\beta - \beta^*\| < \epsilon$, there exists $0 < k_0 \leq \infty$ such that $|\tilde{x}_1^\mathsf{T}(\beta_1 - \beta_1^*)|, |\tilde{x}_2^\mathsf{T}(\beta_2 - \beta_2^*)| \leq k_0\epsilon$. We can see that

(a) If $|\tilde{x}_1^\mathsf{T}\beta_1^*| \leq k_0\epsilon$ or $|\tilde{x}_2^\mathsf{T}\beta_2^*| \leq k_0\epsilon$,
$$\mathbf{1}\{|\tilde{x}_1^\mathsf{T}\beta_1^*| \text{ or } |\tilde{x}_2^\mathsf{T}\beta_2^*| \leq k_0\epsilon\} = 1 \geq |\mathbf{1}\{\tilde{x}_1^\mathsf{T}\beta_1 > 0\} - \mathbf{1}\{\tilde{x}_1^\mathsf{T}\beta_1^* > 0\}|,$$
$$|\mathbf{1}\{\tilde{x}_2^\mathsf{T}\beta_2 > 0\} - \mathbf{1}\{\tilde{x}_2^\mathsf{T}\beta_2^* > 0\}|,$$
$$|\mathbf{1}\{\tilde{x}_1^\mathsf{T}\beta_1, \tilde{x}_1^\mathsf{T}\beta_1 > 0\} - \mathbf{1}\{\tilde{x}_2^\mathsf{T}\beta_2^*, \tilde{x}_2^\mathsf{T}\beta_2 > 0\}|.$$

(b) If $|\tilde{x}_1^\mathsf{T}\beta_1^*|, |\tilde{x}_2^\mathsf{T}\beta_2^*| > k_0\epsilon$,
$$\mathbf{1}\{|\tilde{x}_1^\mathsf{T}\beta_1^*| \text{ or } |\tilde{x}_2^\mathsf{T}\beta_2^*| \leq k_0\epsilon\} = 0 \geq |\mathbf{1}\{\tilde{x}_1^\mathsf{T}\beta_1 > 0\} - \mathbf{1}\{\tilde{x}_1^\mathsf{T}\beta_1^* > 0\}|,$$
$$|\mathbf{1}\{\tilde{x}_2^\mathsf{T}\beta_2 > 0\} - \mathbf{1}\{\tilde{x}_2^\mathsf{T}\beta_2^* > 0\}|,$$
$$|\mathbf{1}\{\tilde{x}_1^\mathsf{T}\beta_1, \tilde{x}_1^\mathsf{T}\beta_1 > 0\} - \mathbf{1}\{\tilde{x}_2^\mathsf{T}\beta_2^*, \tilde{x}_2^\mathsf{T}\beta_2 > 0\}|.$$

Thus, $F := M \cdot \mathbf{1}\{|\tilde{x}_1^\mathsf{T}\beta_1^*| \text{ or } |\tilde{x}_2^\mathsf{T}\beta_2^*| \leq k_0\epsilon\}$ is the envelope of $\mathcal{F}_\beta(\bar{x}, \bar{a}, \bar{c}, \bar{s}, \bar{y})$. By assumption 11(ii),

$$\|F\|_{\mathbb{P},2} = M\mathbb{P}\left(|\tilde{x}_1^\mathsf{T}\beta_1^*| \text{ or } |\tilde{x}_2^\mathsf{T}\beta_2^*| \leq k_0\epsilon\right)^{1/2} \leq M(2k_0k_1\epsilon)^{1/2} < \infty$$

for some $0 < k_1 < \infty$.

Since $\mathcal{F}_\beta$ is a class of linear combinations of indicator functions with dimension at most $2^3$, it is VC-subgraph by Lemma 9.6 and Lemma 9.9 in Kosorok (2008). Therefore, its modified bracketing integral $J_{[]}^*(1, \mathcal{F}_\beta)$ is finite.

Now, let

$$\mathbb{G}_n\mathcal{F}_\beta = n^{1/2}[\mathbb{P}_n\{\mathcal{F}_\beta\} - \mathbb{P}\{\mathcal{F}_\beta\}]$$
$$= n^{1/2}\left[\mathbb{P}_n\{\phi_{N;\beta} - \phi_{N;\beta^*} - D(V(\beta) - V(\beta^*))\}\right]$$

be the empirical process indexed by $\beta$. Applying Theorem 11.2 from Kosorok (2008) yields

$$\tau_1 = \mathbb{E}^*\left[n^{1/2}\sup_{\|\beta - \beta^*\| < \epsilon} \cdot |\mathbb{G}_n\mathcal{F}_\beta|\right]/D$$
$$\leq \ell \cdot J_{[]}^*(1, \mathcal{F}_\beta) \cdot \|F\|_{\mathbb{P},2}$$
$$\leq \ell \cdot J_{[]}^*(1, \mathcal{F}_\beta) \cdot M(2k_0k_1\epsilon)^{1/2}$$

for some constant $0 < \ell < \infty$. Hence,

$$\tau_1 \leq c_2 := \ell \cdot J_{[]}^*(1, \mathcal{F}_\beta) \cdot M(2k_0k_1\epsilon)^{1/2} < \infty. \tag{35}$$

(iii) Note that

$$\tau_2 = \mathbb{E}^* \left[ n^{1/2} \sup_{\|\beta - \beta^*\| < \epsilon} \left| \{V(\beta) - V(\beta^*)\} \left\{ 1 - \frac{\mathbb{E}_n\{\phi_D\}}{D} \right\} \right| \right]$$

$$\leq \sup_{\|\beta - \beta^*\| < \epsilon} \left| V(\beta) - V(\beta^*) \right| \cdot \mathbb{E}^* \left[ \left| n^{1/2} \left\{ 1 - \frac{\mathbb{E}_n\{\phi_D\}}{D} \right\} \right| \right] = \mathcal{O}(\epsilon)$$

where the last inequality follows from

$$\left| V(\beta) - V(\beta^*) \right| = \frac{|V(\beta) - V(\beta^*)|}{\|\beta - \beta^*\|} \cdot \|\beta - \beta^*\| \leq \sup_{\|\beta - \beta^*\| < \epsilon} |V'(\beta)| \cdot \epsilon \qquad (36)$$

and $\left| \mathbb{E}^* \left[ \left| n^{1/2} \left\{ 1 - \frac{\mathbb{E}_n\{\phi_D\}}{D} \right\} \right| \right] \right| \leq \sqrt{\operatorname{Var}(\phi_D)}/D$.

(iv) Using equations (35) and (36), the centered process is bounded as follows:

$$\text{LHS of } (34) \leq c_1 \epsilon^{1/2} + o_P(1) + \mathcal{O}(\epsilon) \leq c_3 \phi_n(\epsilon) \qquad (37)$$

It can be verified that the mapping $\epsilon \mapsto \phi_n(\epsilon)/\epsilon^{3/2}$ is a decreasing function and is independent of $n$. Furthermore, by the definition of $\hat{\beta}$ as the maximizer of $\widehat{V}(\beta)$, we have $\sup_\beta(\widehat{V}(\beta) - \widehat{V}(\hat{\beta})) = 0 \leq \mathcal{O}_\mathbb{P}(n^{-2/3})$.

As it follows that

$$n^{2/3} \phi_n(n^{-1/3}) = n^{2/3}(n^{-1/6} + n^{-1/3}) = n^{1/2} + n^{1/3} \leq 2n^{1/2}, \quad \forall n \geq 0,$$

we conclude that $n^{1/3}\|\beta - \beta^*\| = \mathcal{O}_\mathbb{P}(1)$. $\qquad\square$

Now, using these established results, we will prove Theorem 4.

*Proof of Theorem 5.* From the proof of Theorem 4 and the form in the last term of (*), we have

$$n^{1/2}\left(\widehat{V}(\beta^*) - V(\beta^*)\right) \xrightarrow{d} \mathcal{N}\left(0, \mathbb{E}[\psi^2_{V(\pi)}]\right).$$

Notice that

$$n^{1/2}\left(\widehat{V}(\hat{\beta}) - V(\beta^*)\right) = n^{1/2}\left(\widehat{V}(\hat{\beta}) - \widehat{V}(\beta^*)\right) + n^{1/2}\left(\widehat{V}(\beta^*) - V(\beta^*)\right).$$

Thus, it suffices to show that

$$n^{1/2}\left(\widehat{V}(\hat{\beta}) - \widehat{V}(\beta^*)\right) = n^{1/2}\left(\widehat{V}(\hat{\beta}) - \widehat{V}(\beta^*) - \{V(\hat{\beta}) - V(\beta^*)\}\right) + n^{1/2}\left(V(\hat{\beta}) - V(\beta^*)\right) = o_\mathbb{P}(1).$$

By (37) and Lemma 2, let $\epsilon = c_4 n^{-1/3}$ for some $0 < c_4 < \infty$. Then, for sufficiently large $n$, the first term is bounded by:

$$n^{1/2}\left(\widehat{V}(\hat{\beta}) - \widehat{V}(\beta^*) - \{V(\hat{\beta}) - V(\beta^*)\}\right)$$

$$\leq \text{LHS of } (34) = \mathcal{O}_\mathbb{P}(n^{-1/6}) = o_\mathbb{P}(1).$$

By the Taylor expansion of $V(\hat{\beta})$ around $\beta^*$, the second term becomes:

$$n^{1/2}\left(V(\hat{\beta}) - V(\beta^*)\right) = n^{1/2}\left[\frac{1}{2}V''(\beta^*)\|\hat{\beta} - \beta^*\|^2 + o_\mathbb{P}\left(\|\hat{\beta} - \beta^*\|^2\right)\right] \qquad (V'(\beta^*) = 0)$$

$$= n^{1/2}\left[\frac{1}{2}V''(\beta^*)\mathcal{O}_\mathbb{P}(n^{-2/3}) + o_\mathbb{P}(n^{-2/3})\right] \qquad (\text{Lemma 2})$$

$$= \frac{1}{2}V''(\beta^*)\mathcal{O}_\mathbb{P}(n^{-1/6}) = o_\mathbb{P}(1).$$

Hence, the desired result follows.

$\qquad\square$

## A.2 Technical details

### A.2.1 Cross-fitting algorithm

As an alternative to employing Donsker class assumption, sample-splitting or cross-fitting can be used to simplify proofs and ensure theoretical properties of the proposed estimator. We provide a cross-fitting algorithm for computing the MR estimator.

---

**Algorithm 1** Compute $\widehat{V}_{\mathrm{MR}}(\pi)$ via cross-fitting.

---

1: $J$: Pre-specified number of folds.
2: Split the data $\{O_i : i = 1, \cdots, n\}$ into $J$ disjoint folds $F_j$, $j = 1, \cdots, J$.
3: **for** $j$ in $1 \ldots J$ **do**
4: $\quad n_j \leftarrow |F_j|$.
5: $\quad$ Fit nuisance models $\hat{\theta}_{-j}$ only using folds $\{F_i : i \neq j\}$.
6: $\quad$ Compute $\widehat{V}_{\mathrm{MR},-j}(\pi)$ based on $\hat{\theta}_{-j}$.
7: **end for**
8: $\widehat{V}_{\mathrm{MR}}(\pi) \leftarrow \sum_{j=1}^{J} (n_j/n) \widehat{V}_{\mathrm{MR},-j}(\pi)$.

---

### A.2.2 Simulation study

**Data generation** We generated $X_1$ from a continuous uniform distribution over the interval $[-0.3, 0.7]$. $A_k$, $C_k^{\bar{a}_k}$, and $S_k^{\bar{a}_k}$ were generated from logistic models

$$\mathrm{logit}(e_1^1(x_1)) = 0.3 + 0.2x_1, \tag{38}$$

$$\mathrm{logit}(c_1^{a_1}(x_1)) = x_1 + a_1 + \eta_1, \quad \eta_1 = 2, \tag{39}$$

$$\mathrm{logit}(p_1^{a_1}(x_1)) = 5x_1 + 3a_1 + 0.5a_1x_1. \tag{40}$$

The intermediate variable $X_2^{a_1}$ were generated via a normal distribution with mean $\mu_1^{a_1}(x_1) = 0.2 + 0.3x_1 + 1.5a_1 + 0.75a_1x_1$ and standard deviation $1.5$. Potential outcomes of second stage indicators are again generated by logistic models

$$\mathrm{logit}(e_2^{a_11}(\bar{x})) = 0.7 + 0.2x_1 - 0.2x_2 - 0.1x_2^2, \tag{41}$$

$$\mathrm{logit}(c_2^{a_1a_2}(\bar{x})) = -3 + x_1 + x_2 + 0.5a_2 + a_2x_2 + \eta_2, \quad \eta_2 = 3.5, \tag{42}$$

$$\mathrm{logit}(p_2^{a_1a_2}(\bar{x})) = 0.8 - 1.42x_1 + 0.8a_1 - 0.65a_2. \tag{43}$$

Finally, the outcome $Y^{a_1a_2}$ is generated from a normal distribution with mean $\mu_2^{a_1a_2}(\bar{x}) = 2.58 - 1.04x_1 + 1.21a_1 - 0.92a_1x_1 + 2.27x_2 + a_2(1.18 + 3.29a_1 + 3.95x_2)$ and standard deviation $1.5$. The process resulted in approximately $4\%$, $8\%$ of censoring rates and $84\%$, $65\%$ of survival rates in the first and second stage, respectively.

To demonstrate the multiple robustness, we conducted experiments across five model specification scenarios (M1-M5) expected to yield consistency, and one scenario (M6) expected to fail. Nuisance models in each scenario are modeled to achieve the following description.

> M1 : All nuisance models are correctly specified.
> M2 : $p_2, m_{p_2}$ are incorrectly specified.
> M3 : $\varphi_2, m_{p_2}$ are incorrectly specified.
> M4 : $\mu_2^{a_1a_2}$ is incorrectly specified.
> M5 : $\mu_2^{a_1a_2}, m_{\mu_2}$ are incorrectly specified.
> M6 : $\varphi_1, \varphi_2, p_1$ are incorrectly specified.

Specifically in models M2-M6, we deliberately introduced misspecification by removing terms from the correct models (38)-(43) or by using a non-linearly transformed variables. For the conditional outcome models $m_{p_2}$ and $m_{\mu_2}$, we employed generalized additive models when correct specification was intended, and ordinary least squares models without intercept for the misspecified cases.

Table 5: Simulated experiments: average single iteration run time (seconds).

|  | Off-policy evaluation | Off-policy learning |
|---|---|---|
| Experiment 1 | 66 | 162 |
| Experiment 2 | 66 | 268 |

Table 6: MIMIC-III data application: average single iteration run time (seconds).

| Principal value search (1) | Multiply robust principal value search (3) | Doubly robust value search (4) |
|---|---|---|
| 3.84 | 2302 | 2.67 |

**Percentage of correct decision in always-survivors (PCD-AS)**  Lemma 1 provides a foundation to compute the PCD-AS. With $h(X_1; \hat{\pi}, \pi^*) = \mathbb{E}[\mathbf{1}\{\hat{\pi}(\bar{X}) = \pi^*(\bar{X}) \mid X_1, A_1 = \pi_1(X_1), C_1 = 0, S_1 = 1\}]$ which is $L^1(\mathcal{P})$, we have, by Lemma 1,

$$\text{PCD}_{\text{AS}}(\hat{\pi}; \pi) = \mathbb{P}\left(\hat{\pi}(\bar{X}) = \pi(\bar{X}) \mid U = 1111\right) = \mathbb{E}\left[\frac{p_1^0(X_1)m_{p_2}^{00}(X_1)}{\mathbb{E}\left[p_1^0(X_1)m_{p_2}^{00}(X_1)\right]}h(X_1; \hat{\pi}, \pi^*)\right].$$

We used empirical version of this formula with true nuisance models and an independently generated large sample of size 100,000 to compute PCD-AS.

**Additional experiment**  We conducted additional experiment using data generated from different forms of correct nuisance models and values of $\eta_1, \eta_2$. $X_1$ were generated from the same continuous uniform distribution over $[-0.3, 0.7]$. $A_k$, $C_k^{\bar{a}_k}$, and $S_k^{\bar{a}_k}$ were generated from

$$\text{logit}(e_1^1(x_1)) = 0.5 + 0.5x_1^2, \tag{44}$$

$$\text{logit}(c_1^{a_1}(x_1)) = x_1^2 + \eta_1, \quad \eta_1 = 2.5, \tag{45}$$

$$\text{logit}(p_1^{a_1}(x_1)) = 3x_1^2 + 5a_1 - 0.5a_1x_1. \tag{46}$$

The intermediate variable $X_2^{a_1}$ were generated via a normal distribution with mean $\mu_1^{a_1}(x_1) = 0.5 - 0.3x_1^2 + a_1 - 0.5a_1x_1$ and standard deviation 1.5. Potential outcomes of second stage indicators are again generated by logistic models

$$\text{logit}(e_2^{a_1 1}(\bar{x})) = 0.7 - 0.5x_1^2 + 0.5x_2 - 0.1x_2^2, \tag{47}$$

$$\text{logit}(c_2^{a_1 a_2}(\bar{x})) = -3 + x_1 + x_2 + 0.5a_2 + a_2x_2 + \eta_2, \quad \eta_2 = 4, \tag{48}$$

$$\text{logit}(p_2^{a_1 a_2}(\bar{x})) = 0.5 + 2x_1 + x_1x_2 - 0.8a_1 + 0.65a_2. \tag{49}$$

The outcome $Y^{a_1 a_2}$ is generated from a normal distribution with mean $\mu_2^{a_1 a_2}(\bar{x}) = -3 + X_1 + 1.5A_1 - 0.5A_1X_1 + \exp(X_2)/100 + A_2(1.5 + A_1 - 0.5X_2)$ and standard deviation 1.5.

From this setting, the first and second stages exhibited censoring rates of approximately 7% and 13%, respectively, with corresponding survival rates of 85% and 70%. The reduced death rates compared to the previous setting is expected to favor the standard AIPW estimator.

Results from this alternative setting, as presented in Figures 4 and 5, continue to demonstrate multiple robustness and consistency in off-policy learning of the always-survivor-optimal value. The 95% confidence intervals derived from the EIF yielded coverage rates of 95.8% ($n = 2000$) and 94% ($n = 5000$), closely aligning with the nominal value. PCD-AS of the MR estimator converged towards one as the training set size increased, with average values of 0.971 ($n = 2000$) and 0.982 ($n = 5000$), and corresponding standard deviations of 0.027 and 0.016, which is closer to one with less variability than the PCD-AS of the AIPW estimator, averaging 0.964 and 0.973 with standard deviations of 0.028 and 0.017, respectively.

### A.2.3  Preprocessing MIMIC-III data

Our preprocessing steps were initiated based on previously established sepsis data for reinforcement learning from Komorowski et al. (2018). For each patient, we considered a set of baseline covariates,

$X_1$ (age, weight, temperature, glucose, blood urea nitrogen, creatinine, WBC, SOFA score), and a reduced set of covariates, $X_2$ (weight, temperature, glucose, blood urea nitrogen, creatinine, white blood cell count, SOFA score), to inform treatment decisions. The primary outcome $Y$ was the SOFA score at the final time point, where lower scores indicate better patient status. For $k = 1, 2$, the intervention $A_k$ was defined as the application of mechanical ventilation. Survival status $S_k$ was determined by comparing chart time to death time, with $S_k = 0$ indicating death. Censoring $C_k$ was defined as $C_k = 1$ if the chart time was greater than or equal to discharge time, or if survivor information was unobserved.

We focused on the last 48 hours of patient data, dividing this period into three time points: baseline (blocs 6-12), intermediate (blocs 13-19), and final (bloc 20). This structure was designed to ensure all patients were confirmed to have sepsis diagnosis at baseline. For each time point $k$, the covariate vector $X_k$ was extracted from the initial block, while binary indicators $A_k$, $C_k$, and $(1 - S_k)$ were set to 1 if the corresponding event occurred at least once within that time point. We implemented outlier removal based on clinically plausible ranges, specifically for temperature (25-60 degrees celsius), white blood cell count ($\leq 400$), creatinine ($> 0$), and weight ($> 0$). Finally, we restricted our analysis to patients with a baseline SOFA score greater than 8 to ensure a focus on individuals with a clinically severe condition. This preprocessing resulted in a final dataset of 1821 patients. The resulting dataset exhibited censoring rates of 51.6% and survival rates of 98.9%.

### A.2.4  Justification of the identification assumptions in the context of the MIMIC-III

MIMIC-III is a standard dataset for reinforcement learning applications, particularly for Markov Decision Process-type problems after suitable preprocessing (Komorowski et al., 2018). Our reliance on Komorowski et al.'s (2018) preprocessing justifies our assumption of sequential randomization. Additionally, we included all available covariates believed to be pertinent to sepsis patient conditions, aiming to control for confounding as thoroughly as possible.

For the selected subpopulation of sepsis patients, we consider an individual's condition to be largely uninfluenced by others. While patients received vasopressin concurrently with mechanical ventilation, 91% of these patients were treated with less than 0.5 mcg/kg/min of vasopressin, and 77% received less than 0.2 mcg/kg/min. The correlations between the maximum vasopressin dose and mechanical ventilation were 0.24 ($k = 1$) and 0.14 ($k = 2$). Moreover, the correlation between the maximum vasopressin dose and the outcome of interest was $-0.069$. Based on these observations, we assert that causal consistency is a reasonable assumption.

We confirmed the probabilistic monotonicity of the censoring indicator, demonstrating that $\mathbb{P}(C_2 = 1 | A_1 = a_1, A_2 = a_2) \geq \mathbb{P}(C_1 = 1 | A_1 = a_1)$. While the low mortality rate prevented empirical verification of survival indicator monotonicity, mechanical ventilation is widely recognized as a critical intervention designed to prolong survival in acute settings, frequently referred to as a "cornerstone of patient management (Fan et al., 2017)." This inherent purpose provides a strong basis for assuming monotonicity of the treatment effect.

Our analysis confirmed significant overlap in covariate distributions across the different treatment groups, thereby supporting the positivity assumption for the propensity scores. Additionally, employing a flexible classifier, such as a random forest, allowed us to estimate response and survival probabilities that were consistently bounded away from zero. These empirical findings indicate that positivity is upheld, at least probabilistically.

Given the unknown true models in this real-world data, evaluating model fit was limited. We chose flexible models (random forest and generalized additive model) to reduce the risk of misspecification and underfitting. A comparison between our MR estimator and the principal Q-learning estimator (the plug-in version of Equation 1) on both training and testing sets consistently resulted in close values. This is implied when the outcome regression models are correctly specified, the scenario which guarantees the consistency of the proposed estimator.

### A.2.5  Sensitivity analysis of the MIMIC-III application to violations of principal ignorability

Assumption 5 is a strong yet untestable assumption, necessitating an analysis to evaluate the sensitivity of our results to its violation. We propose and present the results of this sensitivity analysis in this section.

We define sensitivity parameters

$$\rho_u^{\bar{a}}(X_1) := \frac{\mathbb{E}[Y^{\bar{a}}\mathbf{1}\{\pi(\bar{X}) = \bar{a}\}|X_1, U = u]}{\mathbb{E}[Y^{\bar{a}}\mathbf{1}\{\pi(\bar{X}) = \bar{a}\}|X_1, U = 1111]}, \quad \bar{a} \in \{0,1\}^2, u \in \{0,1\}^4$$

following a tilting model, and $\lambda(X_1) = \mathbb{P}(U = 0011|X_1, A_1 = 1, C_1 = 0, S_1 = 1) - \mathbb{P}(U = 0101|X_1, A_1 = 1, C_1 = 0, S_1 = 1)$ that additionally controls strata assignment probability.

**Theorem 6.** *Under Assumptions 1, 3, 4, 6-8, the always-survivor value function can be written as*

$$V(\pi) = \frac{\mathbb{E}[p_1^0(X_1)m_{p_2}^{00}(X_1)m_\nu^\pi(X_1)]}{\mathbb{E}[p_1^0(X_1)m_{p_2}^{00}(X_1)]}, \tag{50}$$

*where* $m_\nu^{a_1 a_2}(X_1) = \mathbb{E}[\nu^{a_1 a_2}(X_1)\mathbf{1}\{A_2 = a_2\}|X_1, A_1 = a_1, C_1 = 0, S_1 = 1]$, $\nu_2^{a_1 a_2}(X_1) = \mu_2^{a_1 a_2}(X_1)/\omega_{a_1 a_2}(X_1)$, *and*

$$\omega_{01}(X_1) = \frac{m_{p_2}^{00}(X_1)}{m_{p_2}^{01}(X_1)} + \rho_{0101}^{01}(X_1)\left\{1 - \frac{m_{p_2}^{10}(X_1)}{m_{p_2}^{01}(X_1)}\right\} + \rho_{0111}^{01}(X_1)\frac{m_{p_2}^{10}(X_1) - m_{p_2}^{00}(X_1)}{m_{p_2}^{01}(X_1)},$$

$$\omega_{10}(X_1) = \frac{m_{p_2}^{00}(X_1)}{m_{p_2}^{10}(X_1)} + \rho_{011}^{10}(X_1)\left\{1 - \frac{m_{p_2}^{01}(X_1)}{m_{p_2}^{10}(X_1)}\right\} + \rho_{0111}^{10}(X_1)\frac{m_{p_2}^{01}(X_1) - m_{p_2}^{00}(X_1)}{m_{p_2}^{10}(X_1)},$$

$$\omega_{11}(X_1) = \frac{m_{p_2}^{00}(X_1)}{m_{p_2}^{11}(X_1)} + \rho_{0101}^{11}(X_1)\frac{m_{p_2}^{01}(X_1) - m_{p_2}^{10}(X_1)}{m_{p_2}^{11}(X_1)}$$
$$+ \rho_{0011}^{11}(X_1)\frac{m_{p_2}^{10}(X_1) - m_{p_2}^{01}(X_1)}{m_{p_2}^{11}(X_1)} + \rho_{0001}^{11}(X_1)\left\{1 - \frac{m_{p_2}^{01}(X_1) + \lambda(X_1)}{m_{p_2}^{11}(X_1)}\right\}.$$

*Proof.* Let $\mathcal{U}_{11} = \{0001, 0011, 0101, 1111\}$

$$m_{\mu_2}^{11}(X_1) = \mathbb{E}[\mu_2^{11}(\bar{X})\mathbf{1}\{\pi(\bar{X}) = 1\}|X_1, A_1 = 1, C_1 = 0, S_1 = 1]$$
$$= \mathbb{E}[\mathbb{E}\{Y^{11}\mathbf{1}\{\pi(\bar{X}^1) = 1\}|\bar{X}^1, \bar{A} = \mathbf{1}_2, \bar{C}^{11} = \mathbf{0}_2, \bar{S}^{11} = \mathbf{1}_2\}|X_1, A_1 = 1, C_1^1 = 0, \bar{S}^{11} = \mathbf{1}_2]$$
$$= \mathbb{E}[\mathbb{E}\{Y^{11}\mathbf{1}\{\pi(\bar{X}^1) = 1\}|\bar{X}^1, A_1 = 1, C^1 = 0, \bar{S}^{11} = \mathbf{1}_2\}|X_1, A_1 = 1, C_1^1 = 0, \bar{S}^{11} = \mathbf{1}_2]$$
$$= \mathbb{E}[Y^{11}\mathbf{1}\{\pi(\bar{X}^1) = 1\}|X_1, A_1 = 1, C_1^1 = 0, \bar{S}^{11} = \mathbf{1}_2]$$
$$= \sum_{u \in \mathcal{U}_{11}} \mathbb{E}[Y^{11}\mathbf{1}\{\pi(\bar{X}^1) = 1\}|X_1, U = u]\mathbb{P}(U = u|X_1, A_1 = 1, C_1 = 0, \bar{S}^{11} = \mathbf{1}_2)$$

The second equality follows from Assumptions 1 and 4. The third and fourth equality follows from Assumptions 6 and 8. We show that the probability in the last equation can be written as nuisance models and sensitivity parameters. First, we have

$$\mathbb{P}(U = 0101|X_1, A_1 = 1, C_1 = 0, \bar{S}^{11} = \mathbf{1}_2)$$
$$= \mathbb{P}(S_2^{01} = 1, S_2^{10} = 0|X_1, A_1 = 1, C_1 = 0, \bar{S}^{11} = \mathbf{1}_2)$$
$$= \mathbb{P}(S_2^{01} = 1|X_1, A_1 = 1, C_1 = 0, \bar{S}^{11} = \mathbf{1}_2) - \mathbb{P}(S_2^{10} = 1|X_1, A_1 = 1, C_1 = 0, \bar{S}^{11} = \mathbf{1}_2)$$
$$= \frac{\mathbb{P}(S_2^{01} = 1|X_1, A_1 = 0, C_1 = 0, S_1 = 1) - \mathbb{P}(S_2^{10} = 1|X_1, A_1 = 1, C_1 = 0, S_1 = 1)}{\mathbb{P}(S_2^{11} = 1|X_1, A_1 = 1, C_1 = 0, S_1 = 1)}$$
$$= \{m_{p_2}^{01}(X_1) - m_{p_2}^{10}(X_1)\}/m_{p_2}^{11}(X_1)$$

The second and the third inequality is from Assumption 3. Similarly,

$$\mathbb{P}(U = 0011|X_1, A_1 = 1, C_1 = 0, \bar{S}^{11} = \mathbf{1}_2) = \{m_{p_2}^{10}(X_1) - m_{p_2}^{01}(X_1)\}/m_{p_2}^{11}(X_1),$$
$$\mathbb{P}(U = 0001|X_1, A_1 = 1, C_1 = 0, \bar{S}^{11} = \mathbf{1}_2) = 1 - \{m_{p_2}^{01}(X_1) + \lambda(X_1)\}/m_{p_2}^{11}(X_1).$$

Thus,

$$m_{\mu_2}^{11}(X_1) = \mathbb{E}[Y^{11}\mathbf{1}\{\pi(\bar{X}^1) = 1\}|X_1, U = 1111]\times$$

$$\left\{ \frac{m_{p_2}^{00}(X_1)}{m_{p_2}^{11}(X_1)} + \rho_{0101}^{11}(X_1)\frac{m_{p_2}^{01}(X_1) - m_{p_2}^{10}(X_1)}{m_{p_2}^{11}(X_1)} + \rho_{0011}^{11}(X_1)\frac{m_{p_2}^{10}(X_1) - m_{p_2}^{01}(X_1)}{m_{p_2}^{11}(X_1)} \right.$$

$$\left. + \rho_{0001}^{11}(X_1)\left(1 - \frac{m_{p_2}^{01}(X_1) + \lambda(X_1)}{m_{p_2}^{11}(X_1)}\right)\right\}$$

$$= \omega_{11}(X_1)\mathbb{E}[Y^{11}\mathbf{1}\{\pi(\bar{X}^1) = 1\}|X_1, U = 1111]$$

By the similar process, we can show that

$$\omega_{01}(X_1)\mathbb{E}[Y^{01}\mathbf{1}\{\pi(\bar{X}^0) = 1\}|X_1, U = 1111] = m_{\mu_2}^{01}(X_1),$$
$$\omega_{10}(X_1)\mathbb{E}[Y^{10}\mathbf{1}\{\pi(\bar{X}^1) = 0\}|X_1, U = 1111] = m_{\mu_2}^{10}(X_1).$$

Hence, the desired result follows. $\square$

To simplify our analysis, we treated the sensitivity parameters as unknown constants, setting $\rho_u^{\bar{a}}(X_1) = \rho$, $\lambda(X_1) = \lambda$. We varied $\rho$ from 0.8 to 1.25 and $\lambda$ from $-0.2$ to 0. The negative range for $\lambda$ is motivated by recent evidence indicating that earlier mechanical ventilation may lead to better survival outcomes than later intervention (Kim et al., 2024).

We evaluated the policy using the plug-in version of Equation (50) under varying values of sensitivity parameters. The maximum observed relative error of our estimates, when compared to the previously proposed estimates, was 0.12. This finding suggests that the MR estimator is not sensitive to the violation of principal ignorability.

### A.2.6 Computing resources

The off-policy learning simulation ran on an internal cluster, with each iteration on a single core, 8 GB RAM instance. Other experiments and the MIMIC-III application used a CPU machine with 16 GB RAM. Average single iteration run times are reported in Table 5 and Table 6.

For the first experiment, the total run times were 55 minutes for off-policy evaluation and 135 minutes for off-policy learning. For the additional experiment, off-policy evaluation took a similar amount of time, while off-policy learning required 223.3 minutes. For the MIMIC-III application, a total of 50 iterations of off-policy learning required 3.2 minutes for the principal DTR (1), 1918.3 minutes for the multiply robust principal DTR (3), and 2.2 minutes for the doubly robust DTR (4).

Full research project, including preliminary experiments, required more compute than the experiments reported.

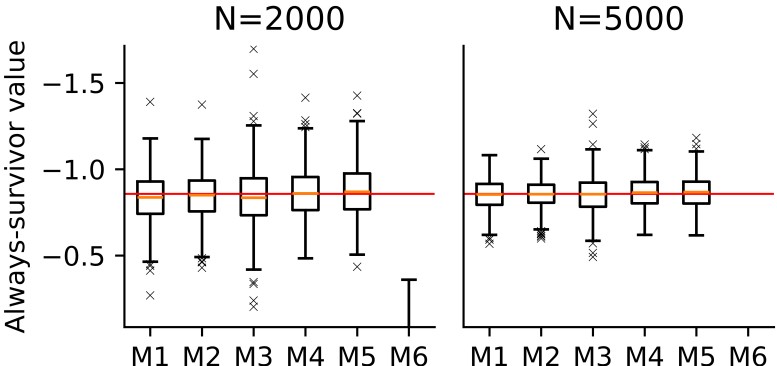

Figure 4: Value estimates under a fixed policy across scenarios M1-M6. The red horizontal line is drawn at the true always-survivor value.

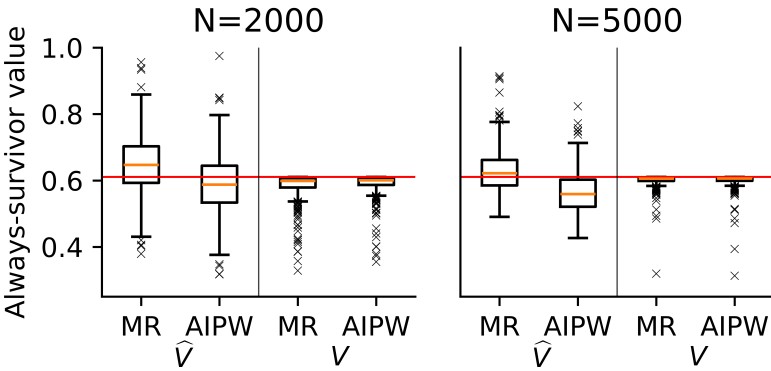

Figure 5: Value estimates from 500 independent simulated off-policy learning runs. Left panels of each plot show $\widehat{V}_{\mathrm{MR}}(\hat{\beta}_{\mathrm{MR}})$ and $\widehat{V}_{\mathrm{AIPW}}(\hat{\beta}_{\mathrm{AIPW}})$, while right panels show $V(\hat{\beta}_{\mathrm{MR}})$ and $V(\hat{\beta}_{\mathrm{AIPW}})$. The red horizontal line indicates the true optimal $V(\beta^*)$.

