# OpenReview forum: "Evaluating and Learning Optimal Dynamic Treatment Regimes under Truncation by Death"
_NeurIPS.cc/2025/Conference — NeurIPS 2025 poster_

### Official Review · Reviewer_SkQo · 2025-06-25

**Clarity:** 3
**Significance:** 3
**Originality:** 3
**Rating:** 4
**Confidence:** 3

**Summary:**

The paper suggests a theoretical framework for dynamic treatment regime (DTR) which allows multistage decisions and handles both truncations by death and censored data. Based on this framework, the authors define the "always survivor" value function, and derive a multiply robust estimator for it. Finally, the authors propose using the value estimator to learn an optimal policy, and test it in a synthetic simulation and a real data scenario extracted from medical health records.

**Questions:**

1. In the discussion, you describe two methods to adapt the resulted policy to the entire population by incorporating the survival rate to the value function. Did you experiment with these methods?
2. Did you analyze the resulted optimal policy over the entire population (what percentage of patients the optimal policy was 00/01/10/11)?
3. How did you evaluate the models fitted from the MIMIC-III data?

**Ethical Concerns:**

["NO or VERY MINOR ethics concerns only"]

**Final Justification:**

The authors addressed my concerns by clarifying the notations and discussing the limitations of the MIMIC-III experiment. While the authors justify assumption 3, I still find it quite limiting. This, along with the softened claims over the MIMIC-III data, I raised my score to borderline accept.

**Limitations:**

The limitations of the paper implied from the assumptions, and further discussed in the appendix

**Paper Formatting Concerns:**

No formatting concerns

**Quality:**

2

**Strengths And Weaknesses:**

## Strengths
1. The method overcomes real-life challenges; multiple decision points, truncation by death and censoring. Overcoming such challenges advances the field and promote real-life applications.
2. The proposed framework and method are well theoretically based, the authors provide proofs in the appendix as well.
3. The authors make a use of real-life data in one of the experiments.

## Weaknesses
1. Clarity of the mathematics; some notations (e.g., $N$, $D$) are not defined, which makes it hard to follow the theoretical results. Since the major contribution of the paper is theoretical, it should be clear enough to follow.
2. Assumption 3 seems inadequate for the medical setting, a treatment may increase the death rate. I think further justifications are needed and/or discuss this limitation and how it affects the MR estimator/AS value function,
3. As far as I understand, the analysis of MIMIC-III data is done over a model that is fitted to the actual data. Hence, I do not understand what these results are justifying; the value is learned over probability models in the same structure the framework considers, which may not mean much in the real-world. The models should be justified first using the raw data, in this case, it is important to discuss the data statistics, as medical health records tend to contain biases which may harm the models. For example, there may be a strong correlation between the treatment decision and the survival probability (physicians may have additional knowledge), or between the censoring probability to the survival probability etc. Without these, it is similar to working with a simulator.

---

> ### Author Rebuttal · Authors · 2025-07-30
>
> We appreciate your insightful and constructive review. Our detailed, point-by-point responses addressing your comments on weaknesses and questions are provided below. We appreciate your understanding regarding our inability to include external links within this rebuttal. A list of references has been provided at the end.
>
> ## Definition of $D$ and $N$
> The terms $D = \mathbb{E}[p_1^0(X_1)m_{p_2}^{00}(X_1)]$ and $N(\pi) = \mathbb{E}[p_1^0(X_1)m_{p_2}^{00}(X_1)m_{\mu_2}^\pi(X_1)]$ are defined as denominator and numerator of Equation 1, respectively. These correspond to $D_K$ and $N_K(\pi)$ when $K=2$ as briefly mentioned in Section 8. We appreciate the reviewers' feedback regarding the omission of a formal introduction for these terms in the main text.
>
> ## Monotonicity assumption
> We recognize that Assumption 3, while standard in principal stratification literature, may not be appropriate in every context. However, its plausibility is high in studies where healthcare providers are precluded from assigning inferior treatments (Sommer and Zeger, 1991; Follmann, 2006). Furthermore, the assumption is automatically satisfied when $S_2^{a_1a_2}=0$ for $a_1+a_2\le1$. It's crucial to note that despite Assumption 3 being imposed, we accommodate scenarios where the outcome, for example, a quality-of-life measure, can worsen following treatment.
>
> ## Details on the MIMIC-III application
> For the selected subpopulation of sepsis patients, we consider an individual's condition to be largely uninfluenced by others. While patients received vasopressin concurrently with mechanical ventilation, 91% of these patients were treated with less than 0.5 mcg/kg/min of vasopressin, and 77% received less than 0.2 mcg/kg/min. The correlations between the maximum vasopressin dose and mechanical ventilation were 0.24 ($k=1$) and 0.14 ($k=2$). Moreover, the correlation between the maximum vasopressin dose and the outcome of interest was −0.069. Based on these observations, we assert that causal consistency is a reasonable assumption.
>
> We confirmed the probabilistic monotonicity of the censoring indicator, demonstrating that $\mathbb{P}(C_2​=1|A_1​=a1​,A_2​=a_2​)\ge \mathbb{P}(C_1​=1|A_1​=a_1​)$. While the low mortality rate prevented empirical verification of survival indicator monotonicity, mechanical ventilation is widely recognized as a critical intervention designed to prolong survival in acute settings, frequently referred to as a "cornerstone of patient management" (Fan et al., 2017). This inherent purpose provides a strong basis for assuming monotonicity of the treatment effect.
>
> MIMIC-III is a standard dataset for reinforcement learning applications, particularly for Markov Decision Process-type problems after suitable preprocessing (Komorowski et al., 2018). Our reliance on Komorowski et al.'s (2018) preprocessing justifies our assumption of sequential randomization. Additionally, we included all available covariates believed to be pertinent to sepsis patient conditions, aiming to control for confounding as thoroughly as possible.
>
> Our analysis confirmed significant overlap in covariate distributions across the different treatment groups, thereby supporting the positivity assumption for the propensity scores. Additionally, employing a flexible classifier, such as a random forest, allowed us to estimate observation and survival probabilities that were consistently near 1. These empirical findings indicate that positivity is upheld, at least probabilistically.
>
> Principal ignorability is a strong yet untestable assumption that has not been justified in previous literature. Thus, it is important to conduct sensitivity analysis to evaluate if the results are sensitive to violation of the assumption. We propose and provide result of a sensitivity analysis in the next section.
>
> Given the unknown true models in this real-world data, evaluating model fit was limited. We chose flexible models (Random Forest and Generalized Additive Model) to reduce the risk of misspecification and underfitting. A comparison between our MR estimator and the principal Q-learning estimator (the plug-in version of Equation 1) on both training and testing sets consistently resulted in close values. This is implied when the outcome regression models are correctly specified, the scenario which guarantees the consistency of the proposed estimator.
>
> Our findings from the MIMIC-III dataset indicate that the derived optimal policy largely favored 00. Treatment assignment was primarily restricted to patients exhibiting either high body weight (exceeding 100 kg) or advanced age (above 55 years). The results obtained using the conventional AIPW estimator were close to our own.
>
> ## Sensitivity analysis
> Under Assumptions 1, 3, 4, and 6-8, we can identify the always-survivor value function as $\mathbb{E}[p_1^0(X_1)m_{p_2}^{00}(X_1)m_{\nu_2}^\pi(X_1)]/D$ with ${\nu_2}^{a_1a_2}(X_1) = {\mu_2}^{a_1a_2}(X_1) / \omega_{a_1a_2}(X_1)$ where $\omega_{00}(X_1) = 1$,
>
> $\omega_{01}(X_1) = \frac{ m_{p_2}^{00}(X_1) }{ m_{p_2}^{01}(X_1) } + \rho_{0101}^{01}(X_1) \left\\{ 1 - \frac{ m_{p_2}^{10}(X_1) }{ m_{p_2}^{01}(X_1) }\right\\} + \rho_{0111}^{01}(X_1) \frac{m_{p_2}^{10}(X_1) - m_{p_2}^{00}(X_1)}{m_{p_2}^{01}(X_1)},$
>
> $\omega_{10}(X_1) = \frac{m_{p_2}^{00}(X_1)}{m_{p_2}^{10}(X_1)} + \rho_{011}^{10}(X_1) \left\\{ 1 - \frac{m_{p_2}^{01}(X_1)}{m_{p_2}^{10}(X_1)}\right\\} + \rho_{0111}^{10}(X_1) \frac{m_{p_2}^{01}(X_1) - m_{p_2}^{00}(X_1)}{m_{p_2}^{10}(X_1)},$
>
> $\omega_{11}(X_1) = \frac{m_{p_2}^{00}(X_1)}{m_{p_2}^{11}(X_1)} + \rho_{0101}^{11}(X_1) \frac{m_{p_2}^{01}(X_1) - m_{p_2}^{10}(X_1)}{m_{p_2}^{11}(X_1)} + \rho_{0011}^{11}(X_1) \frac{m_{p_2}^{10}(X_1) - m_{p_2}^{01}(X_1)}{m_{p_2}^{11}(X_1)} + \rho_{0001}^{11}(X_1) \left\\{1 - \frac{m_{p_2}^{01}(X_1) + \lambda(X_1)}{m_{p_2}^{11}(X_1)} \right\\}.$
>
> Here, $\rho_{u}^{\bar a}(X_1) := \frac{\mathbb{E}[Y^{\bar a} \mathbf{1}\\{\pi(\bar X)=\bar a\\}|X_1,U=u]}{\mathbb{E}[Y^{\bar a}\mathbf{1}\\{\pi(\bar X)=\bar a\\}|X_1,U=1111]}$ and $\lambda(X_1) = \mathbb{P}(U=0011|X_1,A_1=1,C_1=0,S_1=1) - \mathbb{P}(U=0101|X_1,A_1=1,C_1=0,S_1=1)$ are sensitivity parameters. The former is sometimes referred to as a tilting model. Principal ignorability is satisfied if $\rho$’s are equal to one and $\lambda(X_1)=0$.
>
> To simplify our analysis, we treated the sensitivity parameters as unknown constants, setting $\rho_{u}^{\bar a}(X_1) = \rho$, $\lambda(X_1) = \lambda$. We varied $\rho$ from 0.8 to 1.25 and $\lambda$ from -0.2 to 0. The negative range for $\lambda$ is motivated by recent evidence indicating that earlier mechanical ventilation may lead to better survival outcomes than later intervention (Kim et al., 2024)
>
> We evaluated the policy using the new identification under varying values of sensitivity parameters. The maximum observed relative error of our estimates, when compared to the previously proposed estimates, was 0.12. This finding suggests that the MR estimator is not sensitive to the violation of principal ignorability.
>
> ## Adapting the policy in practice
> While experimental validation of these specific methods was not conducted, we anticipate that our proposed approach will perform effectively for data with well-represented always-survivor subgroups. This involves categorizing patients based on their estimated always-survival probability and applying policies. The restricted optimal treatment regime (Zhou et al., 2019) was suggested as an alternative, but exploring this method is beyond the current scope of our work and is reserved for future research.
>
> ## References
> - Sommer, A., & Zeger, S. L. (1991). On estimating efficacy from clinical trials. Statistics in medicine, 10(1), 45-52.
> - Follmann, D. (2006). Augmented designs to assess immune response in vaccine trials. Biometrics, 62(4), 1161-1169.
> - Fan, E., Del Sorbo, L., Goligher, E. C., Hodgson, C. L., Munshi, L., Walkey, A. J., ... & Brochard, L. J. (2017). An official American Thoracic Society/European Society of Intensive Care Medicine/Society of Critical Care Medicine clinical practice guideline: mechanical ventilation in adult patients with acute respiratory distress syndrome. American journal of respiratory and critical care medicine, 195(9), 1253-1263.
> - Komorowski, M., Celi, L. A., Badawi, O., Gordon, A. C., & Faisal, A. A. (2018). The artificial intelligence clinician learns optimal treatment strategies for sepsis in intensive care. Nature medicine, 24(11), 1716-1720.
> - Kim, G., Oh, D. K., Lee, S. Y., Park, M. H., Lim, C. M., & Korean Sepsis Alliance (KSA) investigators. (2024). Impact of the timing of invasive mechanical ventilation in patients with sepsis: a multicenter cohort study. Critical Care, 28(1), 297.

---

> > ### Comment · Reviewer_SkQo · 2025-08-05
> >
> > I thank the authors for the detailed rebuttal. Most of my concerns were addressed, although it is not clear whether the authors will revise the paper accordingly. I do think that a small discussion on the monotonicity assumption would be helpful, as well as the definitions of $D$ and $N$.
> >
> > Regarding MIMIC-III, I am well aware of the dataset, which is why I am concerned. The data processing that results with an MDP (Komorowski et al., 2018) is not ideal, and claims regarding the optimal policy or control-dependent outcomes for sepsis treatment cannot be supported by solving this model, as it is more of a data-driven simulator. I think that a more delicate phrasing, especially when addressing the optimal policy, is required.
> >
> > To conclude for now, I intend to raise the score, but I want to make sure that the authors would revise the discussed issues.

---

> > > ### Author Response · Authors · 2025-08-07
> > >
> > > We appreciate the reviewer's comments, and the manuscript has been revised to address the concerns raised. While we could not attach the updated version with this rebuttal, the revised manuscript now includes explicit definitions for $D$ and $N$, an expanded discussion of the assumptions with illustrative examples, details about the estimator for the $K \ge 3$ case, and the sensitivity analysis.
> > >
> > > Regarding the application of our work to the MIMIC-III dataset, we acknowledge the reviewer's point that the results should not be interpreted as a definitive optimal real-world clinical policy. While we note that the MIMIC-III dataset is a widely adopted standard in causal inference and precision medicine, we have revised Section 7 of our manuscript to reflect the concern.

---

### Official Review · Reviewer_JJLd · 2025-06-29

**Clarity:** 2
**Significance:** 3
**Originality:** 3
**Rating:** 5
**Confidence:** 2

**Summary:**

The authors developed a novel framework for dealing with Dynamic Treatment Regimes (DTR).
Specifically, their approach deals with multi-stage DTR f focusing on the always-survivor value function.
The proposed method is formally proven to be multiply robust against misspecification of the outcome model.
Finally, the authors run both a synthetic experiment and a real word one to demonstrate the effectiveness of their framework.

**Questions:**

1. During you work you used the word censoring as a synonym of right censoring. Is it possible to apply your framework also in presence of left censoring or interval censoring?
2. On line 86 you used the variable A. However, you never formally defined it.
3. Table 2 shows different scenarios in which your framework assures consistency. However, looking at Figure 1, it is possible to see that the sparsity differs in different cases. Is this behavior random or is it an indication that some scenarios are more easily manageable than others?
Misspecification

**Ethical Concerns:**

["NO or VERY MINOR ethics concerns only"]

**Final Justification:**

The authors exhaustively and satisfactorily addressed every point raised during the review. Given that my questions were driven by curiosity rather than critical feedback, I confirm my positive assessment.

**Limitations:**

Yes

**Paper Formatting Concerns:**

No concerns.

**Quality:**

4

**Strengths And Weaknesses:**

## Strengths

The model is capable of dealing with multi-stage DTR even in situation where the outcome is misspecified.
Specifically, the authors presented a robust off-policy learning algorithm to learn the always-survivor value function


## Weaknesses

The High number of assumptions can be a minor issue since they must be verified in order to assure the property of the model on real data.

---

> ### Author Rebuttal · Authors · 2025-07-30
>
> We appreciate your insightful and constructive review. We wish to clarify that the variables $A_k$’s are defined as treatment assignments in lines 72-73, and the bar notation is defined in lines 83-84 of the manuscript. We acknowledge the comment regarding the number of assumptions. We wish to emphasize that the imposed assumptions are standard within both principal stratification and dynamic treatment regimen literature.
>
> Our detailed, point-by-point responses addressing your comments on weaknesses and questions are provided below.
>
> ## Types of censoring
> Our current work focused on a specific type of censoring to simplify the analysis while retaining practical relevance. Although we did not explicitly investigate other forms of censoring, we believe our framework remains applicable. Its core strength lies in handling truncation by death through principal stratification, a mechanism not inherently reliant on the specific type of censorship. Under an alternative censoring mechanism, the primary difference in the estimator would reside in the inverse probability of censoring weighting component, specifically impacting the definition of the nuisance function $\varphi_k()$.
>
> ## Stability of estimator under varying consistency scenarios
> We suppose what the reviewers refer to as sparsity is about the stability of estimators. Should this interpretation be inaccurate, please inform us so that we can elaborate. While not strictly proven, we anticipate the MR estimator will exhibit enhanced stability when the probability models ($p_1, p_2, \varphi_1, \varphi_2$) are correctly specified and their values are bounded sufficiently far from zero.

---

### Official Review · Reviewer_5NiR · 2025-07-01

**Clarity:** 3
**Significance:** 3
**Originality:** 3
**Rating:** 4
**Confidence:** 4

**Summary:**

This paper studies the problem of evaluating and optimizing dynamic treatment regimes (DTRs) from data that are truncated by death, i.e., with patients who die no longer produce new data. To address this challenge, the authors propose a principal stratification method based on a semiparametrically efficient and multiply robust estimator for the always-survivors' value function, i.e., the value function for the individuals who are never truncated by death. Experiments show that the proposed estimator has improvements over the standard AIPW estimator.

**Questions:**

See the weaknesses above.

**Ethical Concerns:**

["NO or VERY MINOR ethics concerns only"]

**Limitations:**

See the weaknesses above.

**Quality:**

4

**Strengths And Weaknesses:**

The problem of truncation by death is ubiquitous in health electronic record data. Being able to evaluate and optimize dynamic treatment regimes in this setting is important for causal inference and many downstream applications. This paper is a good submission with a clear contribution. The authors provide complete assumptions for principal stratification and for identifying the always-survivors' value function. The proposed semiparametrically efficient estimator should be useful in practice. We can see this through the experiments in the article.

There are some weaknesses in this article:

The discussion of related work is not quite sufficient and is a bit disorganized in the first two sections. For example, it is unclear to me why truncation by death is different from censoring. They seem to create the same problem with the data; thus, methods developed for censored data might apply to the problem in the paper.

Is the principal stratification method suitable for multiple decision points? We lose a lot of information if we just estimate the always-survivors' value function. The authors mention the method of pessimism in the last section to address this problem, but it is quite unclear how to do this. This could be a major limitation of the proposed method—it only works well for two decision points. In this sense, this paper doesn't make a large difference from the well-studied static setting with one decision point.

Is the semiparametric efficient estimator really novel? It is basically the same as AIPW. The only difference here is that we are dealing with a two-step expectation. By iterated expectation, we need to apply the AIPW formula twice. Similar formulas have been proposed for evaluating DTRs in the literature.

The multiply-robustness is not very meaningful in practice because it requires four or five nuisance estimators to be specified correctly. It is better to show how the convergence rate of the value function estimator depends on the error of these nuisance estimators and to explain the fast convergence due to the multiplicative structure of the error, as people do in doubly-robust estimation. Multiply-robustness sounds stronger than the doubly-robustness in AIPW, but in fact, it is less robust because the estimation problem is just harder than the standard ATE estimation problem where we apply AIPW.

In the simulation, it is unclear why there is an improvement over AIPW. Could the authors demonstrate improvement for small, medium, and large proportions of always-survivors in the dataset? Could you also compare your proposed method with baseline non-augmented estimators, e.g., stabilized IPW and outcome regression estimators? This is important for demonstrating the efficiency of the proposed estimator.

---

> ### Author Rebuttal · Authors · 2025-07-30
>
> We appreciate your insightful and constructive review. Our detailed, point-by-point responses addressing your comments on weaknesses and questions are provided below.
>
> ## Truncation by death
> For individuals whose observation is subject to truncation by death, the counterfactual outcomes are not merely missing but are, in fact, ill-defined. This fundamental distinction separates such cases from conventional missing data scenarios, as assuming an outcome value is generally inappropriate when it is inherently tied to a mortality event. Furthermore, truncation by death leads to non-comparable treatment groups; individuals who survived under one treatment might have died under a different, counterfactual treatment, thereby rendering the population-level value function ill-defined. This distinction necessitates the development of methods specifically for truncation by death.
>
> ## Principal stratification in $K\ge3$ cases
> With truncation by death, the value function of the outcome collected at the end of the study is undefined. To address this, we should either restrict analysis to the always-survivor stratum, which is the only subgroup for which the final counterfactual outcome is well-defined. Our method still utilizes all available observed data to estimate the always-survivor value. Therefore, despite concentrating on the local average counterfactual outcome, no information is sacrificed.
>
> Provided that Assumptions 3 through 6 are satisfied, principal stratification is a suitable framework for identifying the always-survivor value function. For any $K$, the condition analogous to Assumption 3 is often a natural fit for this problem setting. Similarly, the condition equivalent to Assumption 6 is standard in all multiple-decision DTRs and is inherently satisfied in randomized trials. Assumptions 4 and 5 represent other standard identification assumptions, though their validity necessitates a data-by-data verification.
>
> ## Comparison to the doubly robust (AIPW) estimator
> The proposed estimator fundamentally differs from conventional AIPW, primarily due to the inclusion of the principal score term. Existing doubly robust DTRs cannot handle truncation by death and will often be biased in such scenarios. While our estimator shares structural similarities with doubly robust DTRs—a consequence of their shared semiparametric theoretical foundation—we consider this an advantage, not a drawback. Indeed, it is a strength of our MR estimator that it results in a generalized form of the conventional AIPW estimator, offering broader applicability while addressing the challenge.
>
> Given the fundamental distinction in how truncation by death is handled, a direct comparison between the multiple robustness of our proposed estimator and the double robustness of the conventional AIPW estimator is unjustified, as AIPW cannot account for such truncated outcomes.
>
> Nevertheless, if death is (mis)treated as a censoring event, consistent with the AIPW approach, and we set $p_1(\cdot)=p_2(\cdot)=m_{p_2}(\cdot)= \hat p_1(\cdot)= \hat p_2(\cdot)= \hat m_{p_2}(\cdot)=1$, our estimator aligns with AIPW ((3) from Section 6) and is consistent whenever AIPW is. Therefore, under an interpretation where death is simplified as a censoring event, our estimator indeed provides double robustness. Crucially, our proposed estimator extends this to multiple robustness in more general settings where outcomes truncated by death are rigorously considered undefined.
>
> The observed improvement of the MR estimator over the AIPW estimator in our simulation studies is anticipated, given that the data were specifically simulated to include observations subject to truncation by death. We employed a version of the conventional AIPW estimator for comparison, as it represents a standard approach for death-truncated data in existing literature, and to our knowledge, no other DTR model is currently capable of handling truncation by death.
>
> ## IPW and outcome regression variant of the proposed estimator
> The same simulation was also run with the IPW and Outcome Regression (OR) versions of our proposed estimator. The IPW variant, in addition to survival probability models, includes only inverse probability of censoring weighting (IPCW) components, while the OR variant contains outcome models.
>
> All three estimators (MR, IPW, OR) yielded similar performance. The standard error for the MR estimator (0.029 for N=2000, 0.017 for N=5000) was positioned between the OR estimator (0.022 for N=2000, 0.014 for N=5000) and the IPW estimator (0.032 for N=2000, 0.021 for N=5000). This behavior is theoretically anticipated; under correct model specification, the OR estimator is known to be the most efficient. Our results confirm that the MR estimator achieves comparable efficiency to the OR estimator.

---

### Official Review · Reviewer_XkWQ · 2025-07-01

**Clarity:** 2
**Significance:** 3
**Originality:** 3
**Rating:** 5
**Confidence:** 3

**Summary:**

This paper aims at estimating off-policy evaluation and policy learning in longitudinal settings (DTR settings) where the observed trajectories are truncated by death. In these settings, when a death-event prevents further data measurements, the potential outcomes are undefined, making any population-level treatment effect ill-defined. To overcome this problem, the authors adopt principal stratification and focus on the latent always-survivor stratum, a subgroup which would survive under every treatment combination. They show that under certain assumptions, the expected potential outcome conditional on always-survival (always-survivor value function) is well-defined and identifiable from observational data. Further, they derive a semiparametric efficiency bound and construct a multiply-robust estimator for the treatment always-survivor value function. This estimator is semiparametrically efficient under correct model specifications and multiply-robust to certain model-misspecification. This estimator can be used to find a robust always-survivor-optimal policy. Finally, the authors evaluate their estimator on synthetic and real-world EHR-data in terms of off-policy evaluation and policy learning and show superior performance compared to the AIPW-baseline, which does not explicitly account for death truncations.

**Questions:**

- What would be in illustrative example in practice where we are interested in the potential outcome conditioned on always survivors?
- The AIPW estimator is only suited for censored data, but does not explicitly account for death-truncated data. Is there some systematic relationship between the amount of death-truncated sample points and expected performance gain via MR? Have you investigated how the performance gap between MR and AIPW changes as the death rate varies?
- The authors mention in the discussion, that their results can be extended for $k>=3$ decision points but mention that formalizing this and applying it in practice is complex. How exactly does the MR estimator scale to multiple decision points in practice? Have you tried it? This would be nice to understand the applicability and the claim that their methods generalize beyond single-decision contexts.

**Ethical Concerns:**

["NO or VERY MINOR ethics concerns only"]

**Final Justification:**

My questions have been exhaustively answered in the rebuttal.

**Limitations:**

yes

**Quality:**

3

**Strengths And Weaknesses:**

**Strengths**: The authors address the problem of death-truncation with multiple decision points, a setting that, to the best of my knowledge, has not been formally analyzed before. The paper is mostly written clear and with a well-structured line of argument. The formal analysis is rigorous, presenting relevant assumptions, theorems and proofs. The proposed MR estimator allows for certain model misspecification, which is highly relevant for practical applications, and it works effective for both policy evaluation and optimization. The experimental section is solid: synthetic data with known ground truth illustrate how the estimator behaves under different misspecification scenarios, and the real-world MIMIC-III study confirms the practical gains over a baseline that does not account for death truncation.

**Weaknesses**: The paper lists several assumptions whose practical meaning is not always clear; adding a small, concrete example would help readers see what each assumption means in practice. Finally, the authors claim that they extend beyond single-decision point contexts (previous works). They claim that their approach extends to $k \geq 3$ decision points but show it formally only for $k=2$ and the experiments are also limited to settings with $k=2$. Thus, it is not clear (under which conditions/ at what costs) the approach generalizes beyond the specific two-stage setting.

**Minor weakness/suggestion**: The complex notations and long equation-blocks make the paper hard to follow on first-time reading. Annotating long equations and/ or moving some formal assumptions/ theorems to the appendix leaving only informal versions in the main text might improve readability.  Also, there is a small typo in the caption of section 5.

---

> ### Author Rebuttal · Authors · 2025-07-30
>
> We appreciate your insightful and constructive review. Our detailed, point-by-point responses addressing your comments on weaknesses and questions are provided below.
>
> ## Illustrative example
> Our framework allows the outcome of interest $Y$ to be a measure of quality-of-life (QOL), functional status, or medical expenses. This often-overlooked aspect of survival-related DTR literature is critical, as patient-centered research increasingly prioritizes QOL and functional status as primary outcomes. Beyond clinical efficacy, the integration of expected medical expenses is crucial for cost-effectiveness analyses and health technology assessments.
>
> Additionally, we provide additional interpretations in the context of MIMIC-III for some assumptions:
> - Causal consistency holds if a patient's SOFA score is not affected by other patients’ scores, and no hidden variations or side effects are influencing the score.
> - Monotonicity holds if mechanical ventilation benefits survival.
> - Principal ignorability holds if the always-survivor characteristics match observed survivor characteristics, but not necessarily the opposite.
>
> ## Comparison of the MR and AIPW Estimators
> In our simulation experiments and the MIMIC-III application, the gap in off-policy evaluation did not appear to be directly influenced by the mortality rate. Instead, we speculate this gap widens when the covariate or outcome distributions of the always-survivor stratum significantly diverge from those of the observed survivors. While a low death rate likely increases the overlap between the always-survivor stratum and observed survivors, our proposed estimator can still substantially reduce bias even with low mortality if these two strata exhibit considerable differences.
>
> ## Details about estimators in $K\ge3$ cases
> In the main text, we introduced the identification of the always-survivor value function, which directly leads to a plug-in estimator. The EIF is then derived as follows, with additional notations introduced:
> $\\{ \phi_{N_K(\pi)}(O) - V_{\text{AS},K}(\pi) \phi_{D_K}(O) \\} / D_K(O)$ where
> $$\phi_{D_K}(O)
> 	 = \prod_{k=1}^K \frac{(1-A_k)(1-C_k)}{\varphi_k^{{\bf 0}\_k}(\bar X_k)} S_k
> 	  + \sum_{k=1}^K \left\\{ \prod_{j=0}^{k-1} \frac{(1-A_j)(1-C_j)}{\varphi_j^{{\bf 0}\_j}(\bar X_j)} - \prod_{j=0}^k \frac{(1-A_j)(1-C_j)}{\varphi_j^{{\bf 0}\_j}(\bar X_j)} \right\\} Q_{S,k},$$
>
> $$\phi_{N_K(\pi)}(O)
> 	= Q_{Y,1} \phi_{D_K}(O)
> 	 + \left[ \sum_{k=1}^K \prod_{j=1}^k \frac{\mathbf1\{\pi_j(\bar X_j)=A_j\}(1-C_j)S_j}{\varphi_j^{\pi}(\bar X_j)p_j^{\pi}(\bar X_j)} \big\\{Q_{Y,j+1} - Q_{Y,j}\big\\} \right] Q_{S,1}$$
> with $D_K$, $N_K(\pi)$ defined as in Section 8 of the manuscript, and
> $$Q_{S,K} = S_1\prod_{j=2}^K p_j^{{\bf 0}\_j}(\bar X_j),$$
>
> $$Q_{S,j} = \mathbb{E}[ Q_{S,j+1} | \bar X_j, \bar A_j = {\bf0}_j, \bar C_j = {\bf0}_j], \quad j=1,\dots,K-1,$$
>
> $$Q_{Y,K+1} = Y, \quad Q_{Y,K} = \mu_K^\pi(\bar X),$$
>
> $$Q_{Y,j} = \mathbb{E}[Q_{Y,j+1} | \bar X_j, \bar A_j = \bar\pi_j(\bar X_j), \bar C_j = {\bf0}_j, \bar S_j = {\bf1}_j], \quad j=1,\dots,K.$$
>
> The derivation is technically tedious, yet does not provide information beyond the $K=2$ case, and thus is not presented in the main text.
>
> ## Scalability of the MR estimator in $K\ge3$ cases
> The computational evaluation time and the number of nuisance models for our proposed estimator are comparable to those of the conventional doubly robust (DR) estimator. Specifically, the scalability concerning the nuisance function space mirrors that of the DR estimator. While the single-decision DR estimator generalizes to the $K$-decision case by incorporating two additional nuisance functions (a propensity score and an outcome regression model) for each elongated point, our proposed MR estimator requires three additional nuisance functions (a propensity score, an outcome regression, and a survival probability model) for each elongated point.
>
> In terms of computational requirements, our MR estimator's scalability is also similar to that of the conventional DR estimator. The MR estimator typically requires approximately three times the computation of the DR estimator. This is because the denominator of the MR estimator inherently takes the form of a DR-like formula, with two similar forms also appearing within the numerator.
>
> ## Improvements
> We appreciate the reviewers' comments highlighting areas for improvement. The manuscript has been revised accordingly, and the specific edits are detailed below.
> - We added a table of notations to our manuscript. It summarizes each symbol in Sections 2, 3, and its corresponding description.
> - We organized long equations into a shorter version with additional notations similar to the formula for $K\ge 3$ case.
> - We fixed a typo in Table 3 of Section 5.

---

### Official Review · Reviewer_L5GZ · 2025-07-04

**Clarity:** 4
**Significance:** 3
**Originality:** 3
**Rating:** 5
**Confidence:** 4

**Summary:**

The paper develops a method for learning optimal dynamic treatment regimes (DTR) in the presence of both censoring and truncation by death (survival). Due to the ill-defined nature of potential outcomes after death, the authors focus solely on the population of “always survivors”.  For that, the authors used a principal stratification approach. Additionally, the work relies on the standard assumptions of the time-varying potential outcomes framework, monotonous survival and censoring, and a missing at random mechanism (MAR). Then, to learn the optimal DTR, the authors aimed to maximize an always-survivor policy value with backwards-recursive Q-learning. Furthermore, the paper proposes a semi-parametric efficient estimator for the always-survivor policy value so that the estimation and, thus, the subsequent maximisation are first-order insensitive to the misspecification of the nuisance functions. To demonstrate the efficiency and robustness of their method, the authors conducted several synthetic and real-world experiments

**Questions:**

- What is the definition of $D(O)$ in Theorem 2 (I don’t think it was defined in the main part)?
- I wonder how Table 2 would generalize for $K > 2$? Is there any pattern or a rule for what groups of nuisance functions should be consistently estimated for overall consistency?

**Ethical Concerns:**

["NO or VERY MINOR ethics concerns only"]

**Final Justification:**

The authors have clarified all my questions, and I keep my initial positive score for the paper.

**Limitations:**

The paper has clearly stated all the necessary assumptions required for identification and estimation. Also, where possible, the authors discussed the realism of the necessary assumptions and the impossibility of identification/inference when those assumptions are violated.

Furthermore, the authors provided a discussion on how to extend their method to $K > 2$ time-steps.

**Quality:**

4

**Strengths And Weaknesses:**

The main problem of the paper (= optimal DTR learning for always-survivers) is well-motivated and clearly formulated. Although the authors work with a very specific setting ( = DTR + censoring + truncation by death), it is highly relevant for real-world medical datasets like MIMIC-III. The paper is clearly written, well-structured: it suggests a principled, efficient estimation method that achieves the semi-parametric efficiency bound and possesses multiple robustness (I really like the summarisation of multiple robustness in Table 2).

Regarding "Originality" and "Significance": although the paper follows a standard "blueprint" for deriving efficient estimators, I acknowledge the complexity of the setting and rigorous derivation of accompanying statements (e.g., a second-order remainder required for Theorems 3 and 4).

I have not found any major weaknesses in the paper.

---

> ### Author Rebuttal · Authors · 2025-07-30
>
> We appreciate your insightful and constructive review. Our detailed, point-by-point responses addressing your questions are provided below.
>
> ## Definition of $D$ and $N$
> The terms $D = \mathbb{E}[p_1^0(X_1)m_{p_2}^{00}(X_1)]$ and $N(\pi) = \mathbb{E}[p_1^0(X_1)m_{p_2}^{00}(X_1)m_{\mu_2}^\pi(X_1)]$ are defined as denominator and numerator of Equation 1, respectively. These correspond to $D_K$ and $N_K$ when $K=2$ as briefly mentioned in Section 8. We appreciate the reviewers' feedback regarding the omission of a formal introduction for these terms in the main text.
>
> ## Details about estimators in $K\ge3$ cases
> In the main text, we introduced the identification of the always-survivor value function, which directly leads to a plug-in estimator. The EIF is then derived as follows, with additional notations introduced:
> $\\{ \phi_{N_K(\pi)}(O) - V_{\text{AS},K}(\pi) \phi_{D_K}(O) \\} / D_K(O)$ where
> $$\phi_{D_K}(O)
> 	 = \prod_{k=1}^K \frac{(1-A_k)(1-C_k)}{\varphi_k^{{\bf 0}\_k}(\bar X_k)} S_k
> 	  + \sum_{k=1}^K \left\\{ \prod_{j=0}^{k-1} \frac{(1-A_j)(1-C_j)}{\varphi_j^{{\bf 0}\_j}(\bar X_j)} - \prod_{j=0}^k \frac{(1-A_j)(1-C_j)}{\varphi_j^{{\bf 0}\_j}(\bar X_j)} \right\\} Q_{S,k},$$
>
> $$\phi_{N_K(\pi)}(O)
> 	= Q_{Y,1} \phi_{D_K}(O)
> 	 + \left[ \sum_{k=1}^K \prod_{j=1}^k \frac{\mathbf1\{\pi_j(\bar X_j)=A_j\}(1-C_j)S_j}{\varphi_j^{\pi}(\bar X_j)p_j^{\pi}(\bar X_j)} \big\\{Q_{Y,j+1} - Q_{Y,j}\big\\} \right] Q_{S,1}$$
> with $D_K$, $N_K(\pi)$ defined as in Section 8 of the manuscript, and
> $$Q_{S,K} = S_1\prod_{j=2}^K p_j^{{\bf 0}\_j}(\bar X_j),$$
>
> $$Q_{S,j} = \mathbb{E}[ Q_{S,j+1} | \bar X_j, \bar A_j = {\bf0}_j, \bar C_j = {\bf0}_j], \quad j=1,\dots,K-1,$$
>
> $$Q_{Y,K+1} = Y, \quad Q_{Y,K} = \mu_K^\pi(\bar X),$$
>
> $$Q_{Y,j} = \mathbb{E}[Q_{Y,j+1} | \bar X_j, \bar A_j = \bar\pi_j(\bar X_j), \bar C_j = {\bf0}_j, \bar S_j = {\bf1}_j], \quad j=1,\dots,K.$$
>
> The derivation is technically tedious, yet does not provide information beyond the $K=2$ case, and thus is not presented in the main text.
>
> Regarding the multiple robustness scenario, one illustrative grouping of nuisance functions involves partitioning them into a set of inverse probability of censoring models (specifically, $\varphi_k()$) and a set of outcome models ($p_k(), m_{p_2}(), m_{\mu_2}()$). For this particular grouping, the MR estimator requires only one of these groups to be correctly specified. This property also holds for the MR estimator when $K=2$. This specific arrangement, among other potential groupings, demonstrates the group-doubly robust nature.
>
> For a more generalized analysis of robustness patterns, assessing the consistency of the numerator and denominator separately proves beneficial, similar to the approach in the proof of Theorem 3. The example grouping for $K=2$ is a direct application of this proof. Specifically, by assuming condition (i-3) on page 16, as demonstrated in the example, consistency requires either condition (i-1) or (ii-3) to hold.

---

> ### Comment · Reviewer_L5GZ · 2025-08-05
>
> Thank you for the rebuttal! You clarified all my questions, and I keep my initial positive score for the paper.

---

### Note · Authors · 2025-08-12

We wish to express our sincere gratitude to the reviewers for their constructive comments. To our knowledge, this work is the first to introduce a principal stratification framework designed for panel data, enabling the off-policy evaluation and learning of treatment policies despite the challenges posed by truncation by death. While this challenge is prevalent in medicine, existing methods largely focus on time-to-death outcomes. Our methodology addresses this critical gap by allowing quality-of-life measures as the outcome of interest, aligning with the growing emphasis on patient-centered care.

Furthermore, our contributions include the development of a multiply robust estimator and the establishment of its theoretical properties, namely local efficiency and asymptotic normality, under standard conditions. These theoretical findings are empirically confirmed through extensive numerical simulations and an application to the MIMIC-III dataset. We are confident that our contribution offers a significant advance to the field of reinforcement learning, precision medicine, and aligns with the high standards of the NeurIPS community.

As a final remark, we have compiled a comprehensive list of revisions made to the manuscript in response to the discussion.

**1. Foundational and Conceptual Clarifications**
- We provided an expanded discussion of the truncation by death and the practical implications of restricting the analysis to the always-survivor stratum.
- We provided implications of the required assumptions with illustrative examples.
- We explicitly defined $D$ and $N$.
- We organized long equations into easily comprehensible versions.

**2. Technical and Methodological Additions**
- We added a table of notations to our manuscript, summarizing each symbol in Sections 2 and 3 and its corresponding description.
- We provide the EIF and a multiply robust estimator for the $K\ge3$ case.

**3. Empirical Analysis and Interpretation**
- Simulation:
    - We included an efficiency comparison with non-robust estimators.
- MIMIC-III Application:
    - We added justifications for the assumptions.
    - We included a sensitivity analysis for an untestable condition.
    - We included a caution when interpreting the MIMIC-III application results.

**4. Editorial and Miscellaneous**
- We fixed a typo in Table 3 of Section 5.

---

### Decision · Program_Chairs · 2025-09-17

**Decision:**

Accept (poster)

**Comment:**

I concur with the reviewers’ positive view of this paper and recommend acceptance. The work tackles an important problem of DTR evaluation under truncation by death and addresses it nicely. The paper is well-situated in the literature and supported by nice empirics. Some reviewers raised concerns about clarity of notation, assumptions, and extensions beyond the two-stage setting, but the authors have provided clarifications and revisions that address these. I trust these improvements will be incorporated into the final version.